# FROM SHELL TO STRUCTURE: SPHERICAL SHELL DIFFUSION FOR MOLECULAR GEOMETRY GENERATION

## ABSTRACT

Diffusion-based generative models have recently advanced the state of the art in 3D molecular conformation generation, yet most existing methods rely on an isotropic Gaussian prior and unstructured Gaussian noise in Euclidean space. By concentration of measure, such Gaussians place most of their mass on a thin high-dimensional shell, but this shell is a statistical artifact of dimensionality rather than a chemically meaningful scale. As a result, initialization and early dynamics are often mismatched, leading to dispersed trajectories, high entropy, and unstable convergence. We propose **Spherical Shell Diffusion (SSD)**, a framework that explicitly replaces the Gaussian prior with a chemically scaled spherical-shell initialization and augments the reverse process with a structured dynamics field combining inward radial attraction, short-range repulsion, and an SE(3)-equivariant correction. This design avoids wasted radial drift, stabilizes early trajectories, and yields denoising processes that better align with molecular geometry. Empirical results on GEOM-Drugs and GEOM-QM9 show that SSD consistently improves both *quality* and *diversity* across multiple diffusion backbones, underscoring the value of combining structured geometric priors with geometry-aware dynamics for 3D molecular generation.

## 1 INTRODUCTION

Generating accurate 3D molecular structures is a central challenge in computational chemistry, molecular design, and drug discovery (Hawkins, 2017; Kitchen et al., 2004; Schneider, 2018). Traditional conformer generators based on stochastic search or physics-based force fields face efficiency and scalability issues on large, flexible molecules (Hawkins, 2017). Recent deep generative models address these limitations by learning molecular geometry distributions directly from data (Mansimov et al., 2019; Gómez-Bombarelli et al., 2018; Walters & Barzilay, 2020; Polykovskiy et al., 2020), with denoising diffusion models achieving state-of-the-art performance on GEOM-QM9 and GEOM-Drugs (Ramakrishnan et al., 2014; Axelrod & Gomez-Bombarelli, 2022).

Existing approaches fall into two settings: (1) *conditional generation*, which builds 3D structures from molecular graphs (e.g., GeoDiff (Xu et al., 2022), ConfGF (Shi et al., 2021), SubGDiff (Zhang et al., 2024)); and (2) *unconditional refinement*, which denoises noisy conformations without graph input (e.g., EDM (Hoogeboom et al., 2022), MuDiff (Hua et al., 2024), GCDM (Morehead & Cheng, 2024), SemlaFlow (Irwin et al., 2025), FlowMol (Dunn & Koes, 2024)). Despite architectural differences, nearly all coordinate-space methods adopt an isotropic **Gaussian prior** for initialization and unstructured Gaussian noise for dynamics. By concentration of measure, Gaussians place most of their mass on a thin spherical shell at radius $\sigma_T\sqrt{3N}$, a purely dimensional quantity unrelated to chemical scales. As a result, training and inference often begin from mismatched radii, producing high-entropy wandering, unstable early trajectories, and inefficient sampling.

To address this, we propose **Spherical Shell Diffusion (SSD)**, which introduces structure in both initialization and dynamics. On initialization, SSD replaces the Gaussian prior with an explicit spherical shell, ensuring that sampling begins from a chemically meaningful radius rather than a high-dimensional concentration artifact. However, a shell alone is insufficient: without tailored dynamics, atoms may still drift aimlessly along the shell. We therefore couple shell initialization with

*shell-aware dynamics*: an inward radial attraction guiding global motion, a short-range repulsion enforcing a minimum separation $d_{\min}$, and a learned SE(3)-equivariant correction field (Xu et al., 2022; Hoogeboom et al., 2022). This design suppresses long-range Gaussian drift, reduces spatial entropy, and directs trajectories along a geometry-aligned path toward valid conformations. We further introduce a coordinate-space extension of flow matching, **SSD-Flow**, demonstrating that spherical-shell priors and geometry-aware drifts extend naturally beyond diffusion to continuous-time normalizing flows.

Table 1 summarizes coordinate-space design choices. Representative diffusion models adopt Gaussian priors with isotropic noise, whereas SSD provides *structure in both directions*: a spherical-shell prior for initialization and geometry-aware reverse dynamics. Latent-space models (Xu et al., 2023; Dunn & Koes, 2024; Irwin et al., 2025) are excluded since their processes operate in latent space. Figure 1 provides a visual overview.

Table 1: Comparison of Gaussian-based coordinate-space models and SSD. Baselines use Gaussian priors and isotropic noise, whereas SSD introduces a spherical-shell prior and structured dynamics.

| Component | Baseline | SSD (ours) |
|---|---|---|
| Initialization | Gaussian prior | Spherical-shell prior |
| Dynamics | Gaussian noise | Structured dynamics |
| Noise direction | Isotropic | Directional |

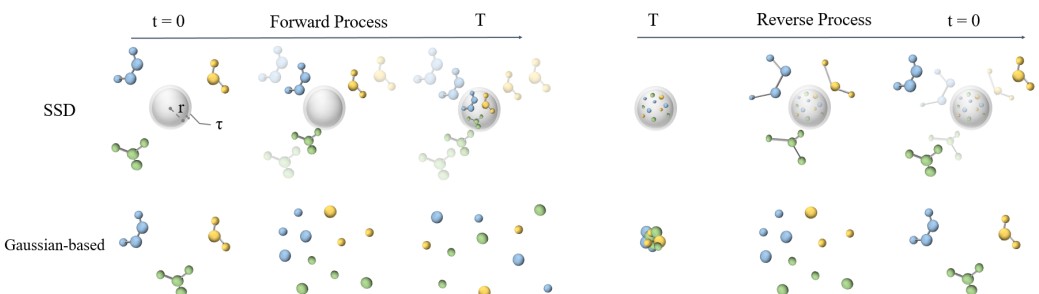

Figure 1: Overview of Spherical Shell Diffusion (SSD). Standard Gaussian diffusion moves states toward a thin high-dimensional concentration shell, but when projected into 3D the reverse-time initialization appears as a small cluster near the origin—a visualization artifact of high-dimensional Gaussians rather than a meaningful molecular radius. In contrast, SSD initializes both the forward and reverse processes on the same chemically meaningful spherical shell, ensuring stable early trajectories and avoiding long-range Gaussian drift. The reverse process then follows a geometry-aligned pathway: global radial contraction draws samples inward, short-range repulsion prevents collapse, and an SE(3)-equivariant correction field refines local structure. Together, these components enable a direct and consistent shell-to-structure generative path.

SSD thus provides a geometry-aware framework broadly applicable across coordinate-based molecular generative models. Our main contributions are as follows. **Unified framework:** SSD couples spherical-shell initialization with shell-aware dynamics, providing a forward–reverse geometric alignment of priors, noise structure, and sampling trajectories. **Improved backbones and results:** SSD consistently improves GeoDiff, SubGDiff, and EDM under identical step budgets, achieving state-of-the-art results (Sec. 4.1) with faster convergence (Sec. 4.3) and stronger quality–diversity trade-offs (Sec. 4.4). **Robustness to larger molecules:** SSD remains stable and accurate on larger, more flexible molecules (Sec. 4.5), demonstrating scalability with system size.

To facilitate reproducibility, we release all code, datasets, and pretrained checkpoints, available at[1].

---

[1]Anonymous repository: `https://github.com/anon-repro/code-release`. A public repo will be released upon acceptance.

## 2 PRELIMINARIES

We represent a molecule as a set of atomic coordinates in 3D space:

$$\mathbf{x}_0 = \{\mathbf{x}_0^{(i)}\}_{i=1}^n \in \mathbb{R}^{n \times 3}, \tag{1}$$

where $n$ is the number of atoms and $\mathbf{x}_0^{(i)} \in \mathbb{R}^3$ is the position of atom $i$. The goal of molecular conformation generation is to recover chemically plausible 3D structures, either *conditionally* from a 2D molecular graph or *unconditionally* from a noisy conformation.

### 2.1 GEOMETRIC AND TASK SETTING.

Molecular conformations are defined only up to rigid-body motions in 3D space, i.e., they are invariant under translations and rotations. This motivates the use of isotropic noise processes and SE(3)-equivariant architectures so that predictions transform consistently under such motions (Xu et al., 2022; Hoogeboom et al., 2022; Zhang et al., 2024; Hua et al., 2024; Morehead & Cheng, 2024). Diffusion-based models fall into two categories: *conditional generation* (2D→3D), where the model lifts a molecular graph into a full 3D conformation, and *unconditional refinement* (3D→3D), where the model receives an existing 3D structure—typically noisy or corrupted—and refines it into a valid conformation. Despite this distinction, both rely on Gaussian-based forward and reverse processes: during training, clean conformations $\mathbf{x}_0$ are gradually perturbed by Gaussian noise to form $\mathbf{x}_t$; during inference, generation is initialized from $\mathbf{x}_T \sim \mathcal{N}(0, \sigma_{\text{gauss}}^2 I)$. By concentration of measure, such Gaussians place most of their mass on a thin high-dimensional shell determined by dimensionality rather than chemistry, motivating our explicit spherical-shell design in SSD.

By concentration of measure, such Gaussians place most of their probability mass on a thin spherical shell of radius $\sqrt{3n}\,\sigma_{\text{gauss}}$ in $3n$-dimensional space (Appendix B; also known as the Gaussian Annulus Theorem (Blum et al., 2020; Vershynin, 2018)). However, this shell is determined solely by dimensionality and variance rather than molecular scales, causing training and inference to begin from unrealistic radii and leading to inefficient, unstable early dynamics. This limitation motivates our **Spherical Shell Diffusion (SSD)**, which makes the shell explicit and introduces geometry-aware dynamics that better align the generative process with molecular structure.

### 2.2 SE(3)-EQUIVARIANCE.

Since molecules have no absolute position or orientation, a valid generative model must respect rigid-body symmetries. The special Euclidean group SE(3) consists of all 3D rotations and translations:

$$g(\mathbf{x}) = R\mathbf{x} + \mathbf{t}, \quad R \in \text{SO}(3), \ \mathbf{t} \in \mathbb{R}^3. \tag{2}$$

A function $f : \mathbb{R}^{n \times 3} \to \mathbb{R}^{n \times 3}$ is *SE(3)-equivariant* if

$$f(g(\mathbf{x})) = g(f(\mathbf{x})), \quad \forall g \in \text{SE(3)}, \tag{3}$$

and *SE(3)-invariant* if $f(g(\mathbf{x})) = f(\mathbf{x})$. Equivariance ensures that rotating or translating a molecule before applying the model yields an output undergoing the same rigid motion. This principle underlies existing SE(3)-equivariant backbones and is preserved by SSD, since all additional drifts depend only on normalized coordinates or pairwise displacements, which transform covariantly under SE(3).

### 2.3 DIFFUSION MODELS.

Denoising diffusion probabilistic models (DDPMs) (Ho et al., 2020) generate samples by reversing a gradual Gaussian noising process. In the continuous-time limit, the forward process is expressed as a stochastic differential equation (SDE) (Song et al., 2021):

$$d\mathbf{x}_t = f(\mathbf{x}_t, t)\, dt + g(t)\, d\mathbf{w}_t, \tag{4}$$

where $f$ is the drift term, $g$ controls the diffusion magnitude, and $\mathbf{w}_t$ is standard Brownian motion. The reverse-time SDE (Song et al., 2021) is

$$d\mathbf{x}_t = \left[ f(\mathbf{x}_t, t) - g(t)^2\, s_\theta(\mathbf{x}_t, t) \right] dt + g(t)\, d\bar{\mathbf{w}}_t, \tag{5}$$

where $s_\theta \approx \nabla_{\mathbf{x}_t} \log p_t(\mathbf{x}_t)$ is a learned score network and $d\bar{\mathbf{w}}_t$ denotes Brownian motion run backward in time. Removing the stochastic term yields a deterministic ordinary differential equation (ODE).

**SSD as a special case.**  In **Spherical Shell Diffusion** (SSD), the forward process differs from Gaussian diffusion by replacing the unconstrained Gaussian perturbation with a geometry-aware drift that keeps samples near a prescribed spherical shell. Concretely, atoms are initialized on a radius-$r$ shell and are guided toward their assigned shell targets by a radial attraction field, while the diffusion coefficient $\sigma_t$ is kept very small. When $\alpha_t = 1$ and $\sigma_t = 0$, trajectories remain exactly on the radius-$r$ sphere, forming a $\delta$-shell. With $\sigma_t > 0$, mild Brownian motion broadens this into a stable *thin spherical shell* of thickness $\tau$, rather than dispersing atoms throughout $\mathbb{R}^{3n}$. Thus, the SSD forward SDE still follows the standard score-based formulation but produces structured thin-shell marginals that align with the geometry of molecular conformations.

## 3   METHODOLOGY

We now describe **Spherical Shell Diffusion (SSD)**. Unlike Gaussian-based diffusion models whose forward and reverse marginals occupy the entire ambient space, SSD introduces geometric structure in both initialization and dynamics. (i) In the forward process, atoms are initialized on a spherical shell and are radially attracted toward assigned shell points; mild isotropic noise broadens the resulting $\delta$-shell into a thin but stable layer of thickness $\tau$ (Appendix G). (ii) In the reverse process, score-based denoising is augmented with a global inward radial attraction term, a short-range repulsion enforcing a minimum separation $d_{\min}$, and an SE(3)-equivariant correction network. These components jointly stabilize trajectories and align the generative process with molecular geometry. SSD-specific parameters ($r$, $d_{\min}$) are computed once from the training split, while $\sigma_t$ is directly inherited from prior diffusion work and is not tuned (Appendix J). To ensure theoretical consistency, we provide in Appendix H a proof that SSD's forward SDE converges to a thin-shell distribution via radial OU dynamics with angular diffusion, and that the reverse SDE satisfies marginal-consistency conditions under score-based time reversal. This justifies the use of shell-based forward and reverse processes in SSD.

**Centering.**  Global translation is removed before diffusion:

$$\mathbf{x}_0 \leftarrow \mathbf{x}_0 - \frac{1}{n} \sum_{i=1}^{n} \mathbf{x}_0^{(i)}. \tag{6}$$

**Shell initialization.**  We sample $n$ points uniformly on a spherical shell of radius $r$:

$$\mathbf{x}_T^{(i)} = r \cdot \frac{\mathbf{v}^{(i)}}{\|\mathbf{v}^{(i)}\|}, \qquad \mathbf{v}^{(i)} \sim \mathcal{N}(0, I), \tag{7}$$

where each $\mathbf{v}^{(i)}$ is normalized onto the shell. Atoms are then assigned shell points via a random permutation $\pi$; shell points are resampled independently for every trajectory to avoid memorization. The radius $r$ is determined once per dataset by computing the mean radius of centered training conformations. This dataset-adaptive "mean shell" provides a stable molecular scale and is shown in Appendix L to outperform small shells, large shells, or Gaussian-derived radii.

**Forward SDE.**  The forward drift is a radial attraction toward each atom's assigned shell point:

$$f^{(i)}(\mathbf{x}_t, t) = \alpha_t \frac{\mathbf{d}_t^{(i)}}{\|\mathbf{d}_t^{(i)}\|}, \qquad \mathbf{d}_t^{(i)} = \mathbf{x}_T^{(\pi(i))} - \mathbf{x}_t^{(i)}. \tag{8}$$

Here $\alpha_t$ follows the backbone schedule, and $\sigma_t$ is a very small shell-noise constant (set to 0 in Flow Matching). The forward SDE is:

$$d\mathbf{x}_t^{(i)} = f^{(i)}(\mathbf{x}_t, t)\, dt + \sigma_t\, d\mathbf{w}_t^{(i)}, \qquad \sigma_t \ll 1. \tag{9}$$

When $\alpha_t = 1$ and $\sigma_t = 0$, samples lie exactly on the radius-$r$ shell; when $\sigma_t > 0$, Brownian motion produces a thin shell of thickness $\tau$ with minimal stochastic support.

**Reverse SDE.**  A key distinction from Gaussian diffusion is that SSD's forward terminal distribution and reverse initial distribution both lie on controlled shells. Thus, the reverse process need

not explore the full Euclidean space; it can follow a direct, geometrically meaningful pathway from the shell to the molecular conformation. To leverage this structure, SSD augments the score-based reverse dynamics with an *inward radial attraction* term:

$$d\mathbf{x}_t^{(i)} = \Big[ -\alpha_t \frac{\mathbf{x}_t^{(i)}}{\|\mathbf{x}_t^{(i)}\|} + \alpha_t \, \mathbf{F}_{\text{rep}}^{(i)}(\mathbf{x}_t) + \alpha_t \, s_\theta(\mathbf{x}_t, t)^{(i)} \Big] dt + \sigma_t \, d\bar{\mathbf{w}}_t^{(i)}. \tag{10}$$

The short-range repulsion is defined as

$$\mathbf{F}_{\text{rep}}^{(i)}(\mathbf{x}) = \sum_{j \neq i} \mathbb{I}[d_{ij} < d_{\min}] \frac{d_{\min} - d_{ij}}{d_{ij}} (\mathbf{x}^{(i)} - \mathbf{x}^{(j)}), \qquad d_{ij} = \|\mathbf{x}^{(i)} - \mathbf{x}^{(j)}\|. \tag{11}$$

Here $d_{\min}$ reflects covalent radii, $\alpha_t$ matches the forward schedule, and $\sigma_t$ allows stochasticity when needed. The inward radial attraction provides global geometric guidance; repulsion prevents collapse; and the SE(3)-equivariant correction refines local structure. This decomposition focuses computation on the physically relevant manifold while preserving the flexibility of the learned score network.

**Inference.** At test time, initialization follows the same shell sampling strategy as in training. The stochastic term can be left on to maintain thin-shell diversity or disabled to obtain deterministic trajectories.

**SE(3) equivariance.** SSD employs existing SE(3)-equivariant backbones—GeoDiff or SubGDiff for conditional generation, and EDM for unconditional refinement—without modification. All additional drifts depend only on normalized directions or pairwise displacements, which transform covariantly under SE(3). Thus, SSD preserves equivariance by construction (Appendix H).

## 4 EXPERIMENTS

**Evaluation Settings and Datasets.** Following standard usage in molecular diffusion models, we consider two settings: *2D→3D conditional generation*, where the model predicts a 3D conformation from a molecular graph, and *3D→3D unconditional refinement*, where the model receives an existing (typically noisy) 3D geometry and refines it into a valid conformation. We evaluate SSD on two widely adopted benchmarks, QM9 and Drugs, which together cover a broad range of molecular sizes and geometric complexity. These datasets are used by nearly all prior molecular diffusion models (GeoDiff, SubGDiff, EDM, MuDiff, GCDM, SemlaFlow), enabling direct and fair comparison under identical preprocessing and evaluation protocols. **QM9** contains small organic molecules with high-quality reference conformers, while **Drugs** consists of substantially larger and more flexible drug-like structures. This large difference in molecular scale makes the QM9–Drugs pair the de facto testbed for assessing both accuracy and robustness of 3D generative models. Additional dataset statistics and preprocessing details (identical to prior work) are provided in Appendix E.

**Baselines.** For *conditional generation*, we compare against GeoDiff (Xu et al., 2022), SubGDiff (Zhang et al., 2024), Tor. Diff. Jing et al. (2022), MCF Wang et al. (2024), and ConfGF (Shi et al., 2021) under the **SubGDiff protocol** (same splits and preprocessing). For *unconditional refinement*, we include EDM (Hoogeboom et al., 2022), MuDiff (Hua et al., 2024), and GCDM (Morehead & Cheng, 2024) following the **SemlaFlow protocol** (Irwin et al., 2025). We also report latent/flow methods GeoLDM, SemlaFlow, and FlowMol. Unless stated otherwise, we adhere to the official training/evaluation settings; extended baseline details are in Appendix A.

**Evaluation Metrics.** We use standard metrics for both settings. Table 2 summarizes these metrics and their optimization directions. **COV-R/P** measures diversity as the fraction of conformers that can be matched within a fixed RMSD cutoff (0.5 Å on QM9, 1.25 Å on Drugs, following SubGDiff). **MAT-R/P** report the corresponding RMSD values as a measure of quality. For unconditional refinement, **Atom/Molecule Stability** follows SemlaFlow (Irwin et al., 2025), **Validity** is checked by RDKit sanitization, and **Uniqueness** measures the percentage of distinct conformations. Full definitions appear in Appendix D.

**Computational Cost.** SSD introduces no additional parameters for coordinate diffusion backbones, leaving both training and inference time unchanged (SubGDiff → SSD-SubGDiff). For latent-flow architectures, SSD-Flow employs a more compact EDM-style encoder than the original SemlaFlow model, reducing computational cost while keeping the flow formulation and number of function evaluations identical. A full parameter and wall-clock comparison is provided in Appendix L.

Table 2: Evaluation metrics grouped by primary focus. ↑: higher is better, ↓: lower is better. All RMSD values are reported in Å (Ångström, 1 Å = 0.1 nm). For conditional generation, COV/MAT use RMSD thresholds of 0.5 Å on QM9 and 1.25 Å on Drugs (SubGDiff protocol Zhang et al. (2024)). For unconditional refinement, atom/molecule stability follows SemlaFlow Irwin et al. (2025), validity is checked by RDKit sanitization, and uniqueness is computed after removing duplicates.

| Metric | Category | Setting | Direction |
|---|---|---|---|
| COV-R (%) ↑ | Diversity | Conditional | Higher is better |
| MAT-R (Å) ↓ | Quality | Conditional | Lower is better |
| COV-P (%) ↑ | Diversity | Conditional | Higher is better |
| MAT-P (Å) ↓ | Quality | Conditional | Lower is better |
| Atom stability (%) ↑ | Quality | Unconditional | Higher is better |
| Mol stability (%) ↑ | Quality | Unconditional | Higher is better |
| Validity (%) ↑ | Quality | Unconditional | Higher is better |
| Uniqueness (%) ↑ | Diversity | Unconditional | Higher is better |

**Training Settings.** We build on the official implementations of GeoDiff, SubGDiff (conditional), and EDM (unconditional). Our SSD variants replace the *Gaussian prior* with a spherical-shell prior and apply structured forward/reverse dynamics, serving as a **drop-in, coordinate-model-grained** module. We further introduce **SSD-Flow**, which performs Flow Matching directly in coordinate space with EDM-style corruption while borrowing SemlaFlow-inspired substructure encodings to maintain compatibility with existing refinement protocols. All results follow a unified protocol, where sampling steps denote diffusion steps (or ODE evaluations) for diffusion models and NFEs for flow-matching models. Detailed implementation settings, including SSD-specific parameters, are provided in Appendix J.

Table 3: Results on QM9 and Drugs for conditional generation (SubGDiff protocol Zhang et al. (2024)). "Steps" denotes the number of *sampling steps* under our unified protocol (diffusion steps or ODE evaluations for diffusion; NFEs for flow matching). ↑ higher is better, ↓ lower is better. Dagger (†) indicates SSD variants built on top of the corresponding backbone (GeoDiff → SSD-GeoDiff, SubGDiff → SSD-SubGDiff). **Bold** marks the best across methods. Entries not reported under Zhang et al. (2024) are shown as '–'. For fairness, we standardize all methods to the SubGDiff evaluation protocol, as the original implementations of Tor. Diff. and MCF use different evaluation settings on the same datasets.

| Dataset | Model | Steps | COV-R ↑ | | COV-P ↑ | | MAT-R ↓ | | MAT-P ↓ | |
|---|---|---|---|---|---|---|---|---|---|---|
| | | | mean | median | mean | median | mean | median | mean | median |
| QM9 | CGCF | – | 78.05 | 82.48 | 36.49 | 33.57 | 0.4219 | 0.3900 | 0.6615 | 0.6427 |
| | ConfVAE | – | 77.84 | 88.20 | 38.02 | 34.67 | 0.4154 | 0.3739 | 0.6215 | 0.6091 |
| | GeoMol | – | 71.26 | 72.00 | – | – | 0.3731 | 0.3731 | – | – |
| | GeoDiff | 5000 | 80.36 | 83.82 | 53.66 | 50.85 | 0.2820 | 0.2799 | 0.6673 | 0.4214 |
| | Tor. Diff. | 5000 | 84.26 | 84.75 | 54.91 | 53.47 | 0.2847 | 0.2803 | 0.6582 | 0.4203 |
| | MCF | 5000 | 87.14 | 88.36 | 55.83 | 54.68 | 0.2649 | 0.2711 | 0.6493 | 0.4538 |
| | SubGDiff | 5000 | 90.91 | 95.59 | 50.16 | 48.01 | 0.2460 | 0.2351 | 0.6114 | 0.4791 |
| | SSD-GeoDiff† | 5000 | 92.00† | 93.80† | 61.40† | 59.75† | 0.2425† | 0.2508† | 0.4188† | 0.4136† |
| | SSD-SubGDiff† | 5000 | **93.20†** | **96.10†** | **63.50†** | **61.80†** | **0.2380†** | **0.2300†** | **0.4100†** | **0.4050†** |
| Drugs | CGCF | – | 53.96 | 57.06 | 21.68 | 13.72 | 1.2487 | 1.2247 | 1.8571 | 1.8066 |
| | ConfVAE | – | 55.20 | 59.43 | 22.96 | 14.05 | 1.2380 | 1.1417 | 1.8287 | 1.8159 |
| | GeoMol | – | 67.16 | 71.71 | – | – | 1.0875 | 1.0586 | – | – |
| | GeoDiff | 500 | 64.12 | 75.56 | 52.79 | 50.29 | – | – | – | – |
| | Tor. Diff. | 500 | 72.10 | 77.78 | 66.88 | 64.41 | 1.0532 | 1.0364 | 1.4725 | 1.4429 |
| | MCF | 500 | 74.45 | 81.33 | 64.97 | 62.29 | 1.0425 | 1.0267 | 1.5284 | 1.4905 |
| | SubGDiff | 500 | 76.16 | 86.43 | – | – | 1.0003 | 0.9905 | – | – |
| | SSD-GeoDiff | 500 | 91.69† | 94.86† | 70.23† | 68.11† | 0.8463 | 0.8247 | 1.1543 | 1.0894 |
| | SSD-SubGDiff | 500 | **92.50†** | **95.80†** | **71.10** | **69.00** | **0.8300†** | **0.8100†** | **1.1400** | **1.0700** |

Table 4: Results on QM9 and Drugs for unconditional refinement (SemlaFlow protocol Irwin et al. (2025)). "Step" denotes the number of *sampling steps* under our unified protocol (diffusion steps or ODE evaluations for diffusion; NFEs for flow matching). ↑ higher is better. Dagger (†) indicates SSD variants built on top of the corresponding backbone (EDM → SSD-EDM, SemlaFlow → SSD-Flow). **Bold** marks the best across methods. Entries not reported in Irwin et al. (2025) are shown as '–'.

| Dataset | Model | Step | Atom Stab↑ | Mol Stab↑ | Valid↑ | Unique↑ |
|---------|-------|------|-----------|-----------|--------|---------|
| QM9 | GCDM | 1000 | 98.7 | 85.7 | 94.8 | 98.4 |
| | EQGAT-diff | 500 | **99.9** | 98.7 | 99.0 | **100.0** |
| | FlowMol | 100 | 99.7 | 96.2 | 97.3 | – |
| | GeoLDM | 1000 | 98.9 | 89.4 | 93.8 | 98.8 |
| | MuDiff | 1000 | 98.8 | 89.9 | 95.3 | 99.1 |
| | EDM | 1000 | 98.7 | 82.0 | 91.9 | 98.9 |
| | SemlaFlow | 100 | **99.9** | 99.7 | 99.4 | 95.4 |
| | SSD-EDM | 1000 | 99.3† | 99.8† | **100.0†** | **100.0†** |
| | SSD-Flow | 100 | **99.9** | **99.9†** | **100.0†** | **100.0†** |
| Drugs | GCDM | 1000 | 89.0 | 5.2 | – | – |
| | EQGAT-diff | 500 | **99.8** | 93.4 | 94.6 | **100.0** |
| | FlowMol | 100 | 99.0 | 67.5 | 51.2 | – |
| | GeoLDM | 1000 | 98.9 | 61.5 | **99.3** | – |
| | MuDiff | 1000 | 84.0 | 60.9 | 98.9 | – |
| | EDM | 1000 | 81.3 | – | – | – |
| | SemlaFlow | 100 | **99.8** | 97.3 | 93.9 | **100.0** |
| | SSD-EDM | 1000 | 90.5† | 70.1 | 98.5 | 99.2 |
| | SSD-Flow | 100 | **99.8** | **97.6†** | **100.0†** | **100.0** |

## 4.1 MAIN RESULTS

We evaluate SSD on representative diffusion backbones: GeoDiff (Xu et al., 2022) and Sub-GDiff (Zhang et al., 2024) for conditional generation, and EDM (Hoogeboom et al., 2022) for unconditional refinement. As shown in Tables 3–4, integrating SSD consistently improves performance under the same sampling budget. In conditional generation, SSD-SubGDiff achieves the strongest results on both QM9 and Drugs. In unconditional refinement, SSD-EDM substantially improves stability and validity. We further evaluate SSD-Flow, which integrates SSD with Flow Matching (Lipman et al., 2023) and SemlaFlow-inspired substructure features (Irwin et al., 2025). We additionally test SSD on the Conformer Fields (MCF) backbone (Wang et al., 2024) under its original GEOM-QM9 protocol, where SSD-MCF improves all four GEOM metrics under both DDPM-1000 and DDIM-50 samplers; full results are provided in Appendix K. This confirms that SSD's gains hold across evaluation protocols. Overall, SSD is coordinate-model-agnostic: it consistently improves both conditional and unconditional backbones under identical sampling budgets, without changing their architectures or loss formulations.

## 4.2 ABLATION STUDY

A pure Gaussian initialization cannot be paired with SSD's radial attraction field, because the reverse radial term assumes a chemically meaningful forward radius, whereas a Gaussian prior concentrates at $r_{\text{Gauss}} = \sigma_T \sqrt{3N}$ purely due to dimensionality. This radius has no molecular interpretation and leads to radial mismatch under SSD dynamics. For this reason, the only meaningful Gaussian-based control is a *Gaussian shell* placed at this concentration radius rather than directly applying SSD dynamics to a raw Gaussian initialization. For consistency, all metrics reported in Sec. 4.2–4.4 use the standard mean-based COV-R and MAT-R measures.

To disentangle initialization from dynamics, we evaluate the five variants in Table 5. (1) The Gaussian baseline uses standard SubGDiff dynamics. (2) Adding repulsion and equivariant correction on top of a Gaussian prior yields only marginal gains, showing that guidance terms alone cannot resolve the radial mismatch. (3) Converting the Gaussian prior into its corresponding Gaussian shell at $r_{\text{Gauss}}$ produces results nearly identical to the baseline—unsurprising, since this shell simply compresses the high-dimensional Gaussian concentration radius into a 3D scalar and carries no chemical meaning. (4) Applying *full SSD dynamics* on this Gaussian shell yields modest improvement but remains far below SSD, because the radius is not aligned with molecular scale. (5) In contrast, full

SSD—combining the dataset-adaptive chemical shell with shell-consistent dynamics—achieves the strongest performance. These ablations, together with the additional analyses in Appendix L, highlight four key principles: radii must match molecular scale; geometric priors (e.g., $d_{\min}$, $\sigma_t$) help only when paired with a shell-aligned forward process; thin-shell noise provides minimal stochastic support rather than a tunable variance parameter; and neither Gaussian variance reduction nor deterministic assignment reproduces SSD's gains. Overall, only SSD aligns initialization, noise structure, and reverse dynamics within the same geometry, yielding substantial and consistent improvements.

Table 5: (Supplemented from rebuttal Table R2) Ablation of initialization and dynamics on GEOM-QM9. The Gaussian concentration shell ($r_{\text{Gauss}} = \sigma_T \sqrt{3 N_{\text{med}}}$) is the only valid Gaussian-based control compatible with SSD's radial contraction.

| Variant | Initialization / Prior | Dynamics | COV-R ↑ | MAT-R ↓ |
|---|---|---|---|---|
| (1) Gaussian baseline | Gaussian | Standard SubGDiff | 90.91 | 0.2460 |
| (2) Gaussian + guidance | Gaussian | Repulsion + corr. field | 90.95 | 0.2455 |
| (3) Gaussian shell ($r_{\text{Gauss}}$) | Thin Gaussian shell | none | 90.92 | 0.2462 |
| (4) Gaussian shell ($r_{\text{Gauss}}$) | Thin Gaussian shell | Full SSD dynamics | 91.5 | 0.244–0.245 |
| **(5) SSD (ours)** | **Mean chemical shell** | **Full SSD dynamics** | **93.20** | **0.2380** |

## 4.3 FASTER CONVERGENCE WITH SSD

A key advantage of SSD is its ability to converge significantly faster than Gaussian-based baselines. Figures 2 show comparisons on QM9 for both conditional generation and unconditional refinement. For conditional generation, SSD-SubGDiff achieves higher COV-R and lower MAT-R across training iterations compared to SubGDiff (Zhang et al., 2024), indicating that SSD simultaneously improves *diversity* (higher COV-R) and *quality* (lower MAT-R). For unconditional refinement, SSD-Flow reaches near-perfect validity and uniqueness much earlier than SemlaFlow. These results confirm that SSD not only improves final performance but also enables more efficient training across generative paradigms. This improvement aligns with our intuition: compact shell initialization produces clearer denoising gradients than highly dispersed Gaussian noise, leading to faster convergence.

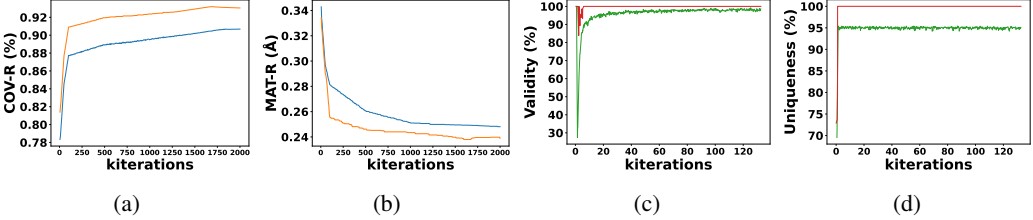

(a)      (b)      (c)      (d)

Figure 2: Faster convergence with SSD on QM9. (Color coding: Orange = SSD-SubGDiff, Blue = SubGDiff, Red = SSD-Flow, Green = SemlaFlow). (a,b) Conditional generation: SSD-SubGDiff vs. SubGDiff, evaluated by COV-R (↑) and MAT-R (↓). (c,d) Unconditional refinement: SSD-Flow vs. SemlaFlow, evaluated by Validity (↑) and Uniqueness (↑). Training curves are reproduced using the official implementations of SubGDiff (Zhang et al., 2024) and SemlaFlow (Irwin et al., 2025). SSD converges faster and consistently outperforms Gaussian-based baselines across both tasks.

## 4.4 ROBUSTNESS ACROSS STEP BUDGETS

Beyond faster convergence during training, SSD also improves sampling efficiency by being more robust to different step budgets. Figures 3 report results on QM9 with sampling steps ranging from 20 to 100. Here, "Steps" denotes diffusion timesteps for score-matching models and the number of ODE function evaluations for flow-matching models (Irwin et al., 2025). Accordingly, SSD-SubGDiff represents our score-matching variant, while SSD-Flow represents the flow-matching variant. For conditional generation, SSD-SubGDiff outperforms SubGDiff (Zhang et al., 2024) by achieving higher COV-R and lower MAT-R across all settings, demonstrating that SSD maintains both greater *diversity* and higher *quality* under limited step budgets. For unconditional refinement,

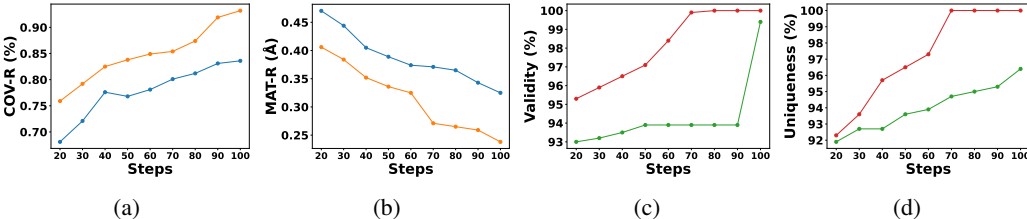

Figure 3: Robust improvements across steps on QM9. (Color coding: Orange = SSD-SubGDiff, Blue = SubGDiff, Red = SSD-Flow, Green = SemlaFlow). "Steps" denotes the number of *sampling steps*, unified across diffusion (steps or ODE evaluations) and flow matching (NFEs). (a,b) Conditional generation: SSD-SubGDiff vs. SubGDiff, evaluated by COV-R (↑) and MAT-R (↓). (c,d) Unconditional refinement: SSD-Flow vs. SemlaFlow, evaluated by Uniqueness (↑) and Validity (↑). All training curves are reproduced using the official implementations of SubGDiff (Zhang et al., 2024) and SemlaFlow (Irwin et al., 2025).

SSD-Flow achieves substantially higher validity and uniqueness than SemlaFlow, even with very few steps, indicating that SSD preserves *quality* and *diversity* in aggressive truncation regimes. These results highlight SSD's robustness to step reduction. This property makes it particularly attractive for efficiency-critical applications such as large-scale virtual screening.

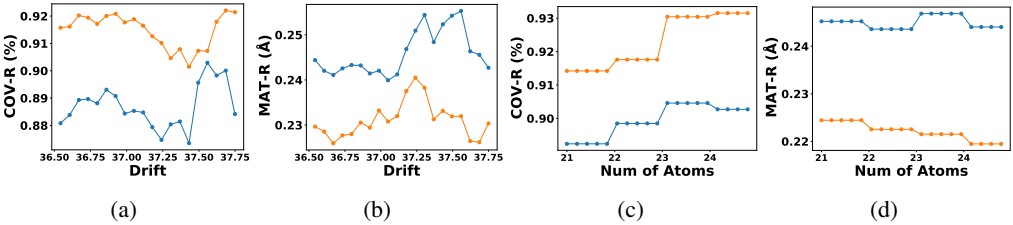

Figure 4: Case study on long-range Gaussian drift. SSD-SubGDiff begins to consistently outperform SubGDiff once drift exceeds ∼36.6 Å (top) or the molecule contains more than 21 atoms (bottom), achieving higher COV-R (↑) and lower MAT-R (↓). (Color coding: Orange = SSD-SubGDiff, Blue = SubGDiff).

### 4.5 CASE STUDY: MITIGATING LONG-RANGE GAUSSIAN DRIFT

Gaussian priors are particularly problematic for larger and more flexible molecules: atoms are often initialized far from chemically plausible regions, which forces large corrective displacements during reverse dynamics and destabilizes refinement. To quantify this effect, we examine COV-R and MAT-R as functions of the total drift (displacement from initialization to the final conformation) and the number of atoms. As shown in Figures 4a–4d, SSD-SubGDiff consistently outperforms SubGDiff once drift exceeds ∼36.6 Å or when molecules contain more than 21 atoms. In these regimes, SSD maintains both higher *quality* (lower MAT-R) and greater *diversity* (higher COV-R), while SubGDiff rapidly deteriorates. To complement these quantitative results, we provide qualitative visualizations in Figures 5 and 6, using conditional generation for direct comparison with reference conformations. The examples confirm that SSD produces stable, chemically plausible structures under both long-drift and large-molecule settings, whereas SubGDiff often yields distorted or unstable intermediates. Overall, these case studies highlight that SSD's robustness arises from the joint effect of shell initialization and structured reverse dynamics, directly addressing the weaknesses of Gaussian-based methods.

**Spatial Entropy of Sampling Trajectories.** To quantify long-range drift in a manner applicable to both SDE- and ODE-based samplers, we measure the spatial entropy of sampling trajectories using two geometry-aware metrics: (i) the radial deviation $\mathbb{E}_t[\,|\,\|x_t\| - r_0\,|\,]$, which captures how far trajectories stray from the chemically meaningful shell radius, and (ii) the path excess ratio

Figure 5: Examples from QM9 under long-drift regimes (conditional generation). SSD-SubGDiff maintains stable geometries, whereas SubGDiff often distorts or produces unstable intermediates.

Figure 6: Examples from QM9 with more than 21 atoms (conditional generation). SSD-SubGDiff produces chemically plausible structures, while SubGDiff frequently yields unstable geometries or misplaces atoms.

PathLen/$\|x_T - x_0\|$, which measures unnecessary wandering relative to the direct path toward the final conformation. Gaussian-based methods exhibit high spatial entropy, with trajectories deviating substantially from the shell and traversing long, meandering paths in $\mathbb{R}^{3N}$. In contrast, SSD maintains consistently low spatial entropy: trajectories stay near the spherical shell and move directly toward the target structure. Full spatial-entropy curves for Gaussian, Gaussian+guidance, and SSD are reported in Appendix L, confirming SSD's suppression of long-range Gaussian drift.

## 5  RELATED WORK

Prior approaches to 3D molecular generation fall into three broad families. (i) Coordinate-space diffusion models such as GeoDiff (Xu et al., 2022), EDM (Hoogeboom et al., 2022), SubGDiff (Zhang et al., 2024), MuDiff (Hua et al., 2024), and GCDM (Morehead & Cheng, 2024) diffuse atomic coordinates with SE(3)-equivariant networks, but all adopt Gaussian priors, leading to dispersed initial states and long denoising trajectories. (ii) Latent- and flow-based models such as GeoLDM (Xu et al., 2023), SemlaFlow (Irwin et al., 2025), and FlowMol (Dunn & Koes, 2024) improve efficiency but still rely on Gaussian priors; our SSD-Flow instead applies Flow Matching directly in coordinate space. (iii) Other distance- or energy-based approaches (e.g., ConfGF (Shi et al., 2021), CGCF (Xu et al., 2021a), GeoMol (Ganea et al., 2021)) emphasize geometric constraints or torsional predictions, which are orthogonal to our focus. A more detailed review of these methods is provided in Appendix A.

## 6  CONCLUSION

We proposed **Spherical Shell Diffusion (SSD)**, a geometry-aware framework for molecular conformation generation. By replacing the Gaussian prior with an explicit spherical shell and coupling it with shell-aware dynamics, SSD avoids wasted radial drift, stabilizes denoising, and preserves SE(3)-equivariance. On QM9 and Drugs, SSD improves accuracy, stability, and diversity across coordinate-based backbones, and extends naturally to Flow Matching through SSD-Flow.

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

## A    SUMMARY OF PRIOR WORK

**Distance- and energy-based methods.**    ConfGF (Shi et al., 2021), CGCF (Xu et al., 2021a), and ConfVAE (Xu et al., 2021b) generate conformations through distance geometry or energy-inspired objectives. By operating on pairwise distances, these approaches naturally preserve roto-translational invariance. However, they follow a two-stage design (distance prediction $\rightarrow$ coordinate reconstruction), where noise in predicted distances may accumulate during reconstruction, sometimes leading to structural inaccuracies.

**Coordinate-space diffusion models.**    GeoDiff (Xu et al., 2022), EDM (Hoogeboom et al., 2022), SubGDiff (Zhang et al., 2024), MuDiff (Hua et al., 2024), and GCDM (Morehead & Cheng, 2024) apply diffusion directly to atomic coordinates using SE(3)-equivariant networks. These models commonly employ Gaussian prior, a simple and widely adopted choice, but one that produces dispersed starting states and long denoising trajectories.

**Latent- and flow-based models.**    GeoLDM (Xu et al., 2023) introduces a latent diffusion model with invariant and equivariant features to improve efficiency and controllability. SemlaFlow (Irwin et al., 2025) adopts a latent-attention architecture with Flow Matching for 3D molecular generation. FlowMol (Dunn & Koes, 2024) develops a flow-matching framework for mixed continuous and categorical variables. Together, these works highlight the flexibility of latent-space modeling and flow-based generative paradigms, offering complementary design perspectives to coordinate-space diffusion.

## B    CONCENTRATION OF THE GAUSSIAN RADIUS

Let $\mathbf{X} \sim \mathcal{N}(\mathbf{0}, \sigma_{\text{gauss}}^2 I_{3n})$ and denote $R = \|\mathbf{X}\|_2$. Then $\frac{R}{\sigma_{\text{gauss}}}$ follows a chi distribution with $k = 3n$ degrees of freedom:

$$\frac{R}{\sigma_{\text{gauss}}} \ \sim \ \chi_k, \qquad k = 3n.$$

Classical properties of the chi distribution yield

$$\mathbb{E}[R] = \sigma_{\text{gauss}}\Big(\sqrt{k} \ - \ \tfrac{1}{2\sqrt{k}} + O(k^{-3/2})\Big), \tag{12}$$

$$\text{Var}(R) = \sigma_{\text{gauss}}^2\Big(\tfrac{1}{2} + O(k^{-1})\Big). \tag{13}$$

In particular, the *typical radius*

$$\rho^\star \ := \ \sigma_{\text{gauss}}\sqrt{k} \ = \ \sigma_{\text{gauss}}\sqrt{3n}$$

approximates $\mathbb{E}[R]$ up to a relative error $O(1/k)$.

**Concentration (thin-shell) inequality.**    The map $f(\mathbf{z}) = \|\mathbf{z}\|_2$ is 1-Lipschitz. By Gaussian concentration,

$$\Pr\Big(\big|R - \mathbb{E}[R]\big| \geq t\Big) \ \leq \ 2\exp\Big(-\tfrac{t^2}{2\sigma_{\text{gauss}}^2}\Big).$$

Equivalently, with $t = \varepsilon\,\rho^\star$,

$$\Pr\Big(\big|R - \rho^\star\big| \geq \varepsilon\,\rho^\star\Big) \ \leq \ 2\exp\Big(-\tfrac{k\varepsilon^2}{2} + O(\varepsilon^2)\Big).$$

Thus with high probability, $R$ lies in the thin spherical shell

$$R \in \big[(1-\varepsilon)\rho^\star, \ (1+\varepsilon)\rho^\star\big],$$

whose relative thickness is $O(1/\sqrt{n})$.

**Connection to SSD.**    For a Gaussian prior $\mathcal{N}(\mathbf{0}, \sigma_{\text{gauss}}^2 I_{3n})$, the typical radius is $\rho^\star = \sigma_{\text{gauss}}\sqrt{3n}$, determined solely by dimension and variance. By contrast, SSD explicitly initializes on a spherical shell of radius $r$ matched to dataset-level molecular scales, and its forward noise only broadens this $\delta$-shell into a thin layer of thickness $\tau$. Hence SSD replaces the *mismatched Gaussian shell* with a *geometry-aware shell* aligned to molecular radii, while structured reverse dynamics further regulate thickness.

**Takeaway.** An isotropic Gaussian in $3n$ dimensions behaves like a random radius tightly concentrated around $\rho^\star = \sigma_{\text{gauss}}\sqrt{3n}$, i.e., a "Gaussian shell." This shell is an artifact of dimensionality, not chemistry, creating a training–inference mismatch. SSD makes the shell explicit and physically meaningful by choosing $r$ to reflect molecular scales and by adding repulsion and thin-shell noise for stability.

## C  DETAILED DESCRIPTIONS OF BASELINE MODELS

**Conditional Generation Models**

- **GeoDiff** (Xu et al., 2022): GeoDiff formulates molecular conformation generation as a denoising diffusion process on atomic coordinates. The model uses an SE(3)-equivariant graph neural network to model the reverse denoising function, ensuring rotational and translational symmetry. Specifically, it implements message passing on invariant features and predicts directional updates via an equivariant edge-to-node transformation. During the forward process, Gaussian noise is added to atomic positions, and the network learns to reverse this process given the molecular graph.

- **ConfGF** (Shi et al., 2021): ConfGF uses conditional Score Matching to model the gradient of the data distribution. Rather than relying on a parameterized diffusion kernel, it directly learns score functions over 3D coordinates using a force field-like design. The model is based on continuous-filter convolution (similar to SchNet (Schütt et al., 2017)) and employs Langevin dynamics to iteratively refine generated structures. It conditions on the molecular graph and guides generation toward conformations that satisfy chemical constraints.

- **SubGDiff** (Zhang et al., 2024): SubGDiff enhances molecular representation learning by introducing substructure-aware diffusion. Instead of injecting noise uniformly across all atoms, it adds noise selectively to subgraphs using a discrete mask distribution. The model also includes three novel training mechanisms: (i) a subgraph prediction loss to guide structure-aware denoising, (ii) expectation state diffusion to simplify the forward process, and (iii) k-step same-subgraph diffusion to improve convergence. Its denoising network uses a standard GNN architecture without explicit geometric equivariance.

- **CGCF** (Xu et al., 2021a): CGCF (Conditional Graph Conformation Flow) combines graph neural networks with flow-based modeling for conditional conformation generation. It conditions on the 2D molecular graph and learns invertible flows that map Gaussian priors to valid 3D conformations, enforcing chemical validity during decoding.

- **ConfVAE** (Xu et al., 2021b): ConfVAE adopts a variational autoencoder framework for conformation generation. It encodes 2D molecular graphs into latent variables and decodes them into 3D conformations using a graph-conditioned decoder. The method emphasizes diversity in conformer sampling via latent variable regularization.

- **GeoMol** (Ganea et al., 2021): GeoMol builds conformations by explicitly predicting torsional angles conditioned on the molecular graph. By combining learned torsional distributions with geometric constraints, GeoMol directly generates chemically valid conformations without iterative refinement.

**Unconditional Refinement Models**

- **EDM (Equivariant Diffusion Model)** (Hoogeboom et al., 2022): EDM applies denoising Score Matching in SE(3)-equivariant latent spaces. It uses the Equivariant Graph Neural Network (EGNN) architecture to perform message passing over both atomic features and coordinates, preserving SE(3) symmetry. EDM perturbs 3D coordinates using Gaussian prior noise and learns to predict score vectors that denoise atom positions. The model operates directly on 3D atom clouds and serves as a standard baseline for unconditional conformation generation.

- **GCDM (Generative Conformation Diffusion Model)** (Morehead & Cheng, 2024): GCDM extends diffusion modeling by simultaneously reconstructing both atomic coordinates and pairwise distance matrices. It uses an SE(3)-equivariant network similar to EGNN, and introduces auxiliary distance-based losses to improve structural realism.

GCDM supports both conditional and unconditional refinement modes through configurable graph encoders.

- **MuDiff** (Hua et al., 2024): MuDiff introduces a variational diffusion framework that supports diverse conformer generation for a given molecule. It models latent variables via amortized variational inference, and performs conformation sampling using conditional Langevin dynamics guided by an energy-based score network. Its score network is based on an SE(3)-equivariant architecture adapted from EGNN.

- **GeoLDM** (Xu et al., 2023): GeoLDM applies latent diffusion modeling to molecular structures. Instead of diffusing directly in coordinate space, it encodes molecules into a lower-dimensional latent space where Gaussian noise is added, and then decodes back into 3D coordinates. This design reduces computational cost while maintaining structural fidelity.

- **EQGAT-diff** (Le et al., 2024): EQGAT-diff employs an equivariant graph attention network for molecular refinement. It incorporates attention-based message passing over both scalar and vector features, enabling efficient modeling of long-range dependencies. The diffusion process follows standard Gaussian corruption and denoising, while the EGAT architecture improves expressiveness.

- **FlowMol** (Dunn & Koes, 2024): FlowMol adopts a flow-matching framework for unconditional molecular conformation generation. It learns a continuous transport map between Gaussian priors and molecular conformations using neural ODEs. FlowMol combines graph-based encoders with equivariant coordinate decoders, achieving fast generation without iterative denoising.

- **SemlaFlow** (Irwin et al., 2025): SemlaFlow is a flow-based generative model that avoids iterative denoising by modeling a bijective mapping between latent and 3D coordinate space. It combines graph message passing and equivariant transformation modules, incorporating multi-head latent attention. Unlike diffusion-based models, SemlaFlow directly generates atomic coordinates, atom types, and charges from latent variables using Flow Matching.

These models form a representative suite spanning diffusion-based, variational, and flow-based generative frameworks. They cover both conditional (2D→3D) and unconditional (3D refinement) tasks, and employ a range of architectures from SE(3)-equivariant GNNs to transformer-based attention networks.

## D    DETAILS OF EVALUATION METRICS

**Summary.**    We follow two established evaluation protocols: (i) for *conditional generation* we adopt the SubGDiff setup (Zhang et al., 2024), and (ii) for *unconditional refinement* we adopt the SemlaFlow setup (Irwin et al., 2025).

Intuitively, **COV** measures *diversity*—the fraction of conformers that can be matched within a fixed RMSD threshold to *any* conformer in the opposite set (reported as recall/precision: COV-R/P); **MAT** measures *quality*—the nearest-neighbor RMSD itself (MAT-R/P). For conditional tasks, we use the thresholds specified in SubGDiff: 0.5 Å on QM9 and 1.25 Å on Drugs. For unconditional tasks, we follow SemlaFlow's definitions: **stability** is based on a 1 Å RMSD cutoff for atomic neighborhoods; **validity** is checked via RDKit sanitization; **uniqueness** is computed after duplicate removal under canonical SMILES. Unless otherwise noted, all distances are reported in Å.

**Conditional Generation Metrics.**    Following the SubGDiff protocol (Zhang et al., 2024), we evaluate generated conformations against ground-truth references using:

- **Coverage (COV-R/P, %, ↑)**: The percentage of reference (R) or predicted (P) conformers that are within the RMSD threshold (0.5 Å on QM9, 1.25 Å on Drugs) of any conformer in the opposite set. Higher indicates better diversity.

- **Matching (MAT-R/P, Å, ↓)**: The RMSD between each reference (R) or predicted (P) conformer and its closest match in the opposite set. Lower indicates higher accuracy.

**Unconditional Refinement Metrics.**    For unconditional refinement, we follow SemlaFlow (Irwin et al., 2025):

- **Atom Stability (%, ↑)**: The percentage of atoms whose positions are within 1 Å RMSD of their ground-truth neighborhoods.
- **Molecule Stability (%, ↑)**: The percentage of molecules in which *all* atoms are stable.
- **Validity (%, ↑)**: The percentage of chemically valid molecules (checked via RDKit sanitization).
- **Uniqueness (%, ↑)**: The percentage of distinct conformations after removing duplicates (via canonical SMILES).

## E    DETAILS OF DATASETS

**GEOM-QM9** (Ramakrishnan et al., 2014) and **GEOM-Drugs** (Axelrod & Gomez-Bombarelli, 2022) are among the most widely adopted datasets for evaluating molecular conformation generation, particularly in both Conditional Generation and Unconditional Refinement settings.

**GEOM-QM9** is a subset of the original QM9 dataset, consisting of small organic molecules with up to 9 heavy atoms (C, O, N, F). Following prior works such as GeoDiff and ConfGF, standard practice selects 40,000 molecules, each with their 5 lowest-energy conformers, resulting in approximately 200,000 training samples. The test set typically contains 200 distinct molecules and around 22,000 total conformers. These settings are designed to balance diversity and computational tractability, while enabling consistent evaluation across models.

**GEOM-Drugs** contains significantly more complex and diverse molecules derived from drug-like compounds. It includes around 430,000 unique molecules, with an average of 44 heavy atoms (up to 181 atoms including hydrogens). Each molecule is annotated with multiple low-energy conformers generated via energy minimization. Most benchmarks use the top 30 conformers per molecule to construct training sets. This dataset poses greater challenges for generative models due to its structural variability and conformational flexibility.

## F    COORDINATE UPDATES IN SE(3)-EQUIVARIANT ARCHITECTURES

SSD is compatible with standard SE(3)-equivariant diffusion backbones. In this work, we instantiate it with GeoDiff/SubGDiff (Xu et al., 2022; Zhang et al., 2024) for conditional generation and EDM (Hoogeboom et al., 2022) (EGNN (Satorras et al., 2021)) for unconditional refinement. Although implementations differ, all backbones share the same principle: coordinate updates are expressed via inter-atomic distances and relative position vectors, ensuring SE(3)-equivariance.

**GeoDiff.**    GeoDiff predicts edge-level scalar gradients in the SE(3)-invariant distance space and maps them back to coordinates via an equivariant transform:

$$\mathbf{g}_t^{(i)} = \sum_{j \neq i} s_{ij} \frac{\mathbf{x}_t^{(i)} - \mathbf{x}_t^{(j)}}{\|\mathbf{x}_t^{(i)} - \mathbf{x}_t^{(j)}\|}, \quad s_{ij} = \frac{\partial \log q}{\partial d_{ij}}, \quad d_{ij} = \|\mathbf{x}_t^{(i)} - \mathbf{x}_t^{(j)}\|.$$

Since the update depends only on normalized relative displacements and distance-invariant scalars, it is equivariant to rotations and translations.

**SubGDiff.**    SubGDiff (Zhang et al., 2024) retains GeoDiff's coordinate update rule and adds subgraph-level masking and prediction modules on top; its SE(3)-equivariance follows directly from GeoDiff.

**EDM (EGNN).**    EDM adopts the Equivariant Graph Neural Network (EGNN) (Satorras et al., 2021), which integrates scalar-to-vector mapping into message passing:

$$\mathbf{x}_t^{(i)} \leftarrow \mathbf{x}_t^{(i)} + \sum_{j \neq i} \left(\mathbf{x}_t^{(i)} - \mathbf{x}_t^{(j)}\right) \phi_x(m_{ij}),$$

where $m_{ij}$ encodes node/edge features and (invariant) distance statistics (e.g., $d_{ij}^2$), and $\phi_x$ outputs a rotation-invariant scalar. This yields the same effect as GeoDiff's scalar-to-gradient chain rule without explicit Jacobian computation. EGNN's SE(3)-equivariance is formally proven in (Satorras et al., 2021).

**SSD with Flow Matching.** Many recent flow-matching methods for molecules operate in latent space (e.g., SemlaFlow (Irwin et al., 2025)). In SSD-Flow, we instead apply Flow Matching directly in coordinate space: we retain SSD's spherical-shell initialization and radial dynamics, and replace score matching with a flow-matching loss (Lipman et al., 2023). For features, we borrow substructure-aware encodings (e.g., subgraph pooling and attention), while coordinate updates remain EGNN-parameterized to preserve SE(3)-equivariance.

**Our usage.** When instantiating SSD with GeoDiff, SubGDiff, or EDM, we leave their coordinate update rules unchanged and simply substitute SSD dynamics for Gaussian dynamics. For Flow Matching, we follow the same coordinate-space formulation while borrowing feature encodings from prior work. Thus, SE(3)-equivariance is inherited from the underlying equivariant networks, while SSD contributes the geometry-aware initialization and dynamics.

## G ANALYSIS OF SHELL THICKNESS

In the forward SDE (Eq. 8), a small constant diffusion coefficient $\sigma_t$ is added to avoid a singular $\delta$-shell. This stochastic term broadens the shell into a thin spherical layer with finite thickness, denoted as $\tau$ in our figures.

Formally, the accumulated radial variance induced by Brownian increments is bounded by

$$\text{Var}[\delta_T] \ \leq \ \int_0^T \sigma_t^2 \, dt. \tag{14}$$

For constant $\sigma_t$, this reduces to

$$\tau_{\max} \ \approx \ \sigma\sqrt{T}, \tag{15}$$

which provides an *upper bound* on the effective shell thickness $\tau$.

In practice, however, the contraction drift in SSD (Eq. 8) acts as a restoring force, suppressing the radial variance. Approximating the dynamics near the shell by an Ornstein–Uhlenbeck (OU) process with drift rate $k$, the stationary variance is

$$\text{Var}[\delta] \ \approx \ \frac{\sigma^2}{2k}, \quad \Rightarrow \quad \tau_{\text{rms}} \ \approx \ \frac{\sigma}{\sqrt{2k}}. \tag{16}$$

Thus, while $\tau_{\max}$ provides a loose upper bound, the effective shell thickness $\tau$ is in fact much smaller, scaling inversely with the contraction strength $k$.

Under standard backbone settings (with $\sigma_t$ chosen as a small constant and $k$ of order one), the resulting $\tau$ is negligible compared to the dataset-specific shell radius $r$ (e.g., several Å on QM9). Hence, $\tau$ primarily serves a numerical role—stabilizing the forward SDE and injecting infinitesimal diversity— without altering the geometry of the initialization.

## H SE(3)-EQUIVARIANCE OF SSD

We prove that SSD is SE(3)-equivariant *modulo global translations*, i.e., on the quotient space $\mathbb{R}^{n \times 3}/\text{Transl}$ standard in molecular modeling and evaluation. Let $g(\mathbf{x}) = R\mathbf{x} + \mathbf{t}$ with $R \in \text{SO}(3)$ act nodewise on $\mathbf{X} = \{\mathbf{x}^{(i)}\}_{i=1}^n$, and let the centering operator be

$$C(\mathbf{X}) \ = \ \mathbf{X} - \mathbf{1}\mu(\mathbf{X})^\top, \qquad \mu(\mathbf{X}) \ = \ \tfrac{1}{n}\sum_{i=1}^n \mathbf{x}^{(i)}.$$

Then $C(g\mathbf{X}) = RC(\mathbf{X})$, i.e., centering removes translations and commutes with rotations. We will show that the centered dynamics satisfy

$$C\big(F(g\mathbf{X})\big) \ = \ R\,C\big(F(\mathbf{X})\big), \tag{17}$$

which establishes SE(3)-equivariance on $\mathbb{R}^{n \times 3}/\text{Transl}$.

**Forward dynamics (centered frame).** Let $\tilde{\mathbf{X}} = C(\mathbf{X})$ and write $\tilde{\mathbf{x}}^{(i)} = \mathbf{x}^{(i)} - \mu(\mathbf{X})$. The SSD forward drift is evaluated in the centered frame as a radial field:

$$\tilde{\mathbf{f}}^{(i)}(\mathbf{X}, t) \;=\; \alpha_t \frac{\tilde{\mathbf{x}}^{(i)}}{\|\tilde{\mathbf{x}}^{(i)}\|}. \tag{18}$$

For any rigid motion $g$, one has $\frac{g(\tilde{\mathbf{x}}^{(i)})}{\|g(\tilde{\mathbf{x}}^{(i)})\|} = R \frac{\tilde{\mathbf{x}}^{(i)}}{\|\tilde{\mathbf{x}}^{(i)}\|}$, since $C(g\mathbf{X}) = RC(\mathbf{X})$ and $\|R\mathbf{v}\| = \|\mathbf{v}\|$. Hence $\tilde{\mathbf{f}}^{(i)}(g\mathbf{X}, t) = R\,\tilde{\mathbf{f}}^{(i)}(\mathbf{X}, t)$. Isotropic Brownian increments are rotation-invariant and translation-invariant; applying $C(\cdot)$ after each infinitesimal step (or equivalently, evaluating the drift in the centered frame) yields

$$C\big(d(g\mathbf{x}_t^{(i)})\big) \;=\; R\,C\big(d\mathbf{x}_t^{(i)}\big),$$

proving Eq. 17 for the forward SDE.

**Reverse dynamics (centered frame).** The reverse SDE combines radial contraction, short-range repulsion, and an SE(3)-equivariant score $s_\theta$ evaluated on centered coordinates:

$$d\mathbf{x}_t^{(i)} \;=\; \Big[ -\alpha_t \frac{\tilde{\mathbf{x}}^{(i)}}{\|\tilde{\mathbf{x}}^{(i)}\|} \;+\; \alpha_t\,\mathbf{F}_{\text{rep}}^{(i)}(\tilde{\mathbf{X}}_t) \;+\; \alpha_t\, s_\theta(\tilde{\mathbf{X}}_t, t)^{(i)} \Big] dt \;+\; \sigma_t\, d\bar{\mathbf{w}}_t^{(i)}. \tag{19}$$

Here $\mathbf{F}_{\text{rep}}^{(i)}$ depends only on pairwise displacements and distances:

$$\mathbf{F}_{\text{rep}}^{(i)}(\tilde{\mathbf{X}}) = \sum_{j \neq i} \mathbb{I}[d_{ij} < d_{\min}] \frac{d_{\min} - d_{ij}}{d_{ij}} \big(\tilde{\mathbf{x}}^{(i)} - \tilde{\mathbf{x}}^{(j)}\big), \quad d_{ij} = \|\tilde{\mathbf{x}}^{(i)} - \tilde{\mathbf{x}}^{(j)}\|.$$

Under $g$, pairwise distances are invariant and relative vectors rotate covariantly; thus $\mathbf{F}_{\text{rep}}^{(i)}(C(g\mathbf{X})) = R\,\mathbf{F}_{\text{rep}}^{(i)}(C(\mathbf{X}))$. By assumption $s_\theta$ is SE(3)-equivariant, so $s_\theta(C(g\mathbf{X}), t) = R\,s_\theta(C(\mathbf{X}), t)$. Combining these with the centered radial term yields

$$C\big(f(g\mathbf{X}, t)\big) \;=\; R\,C\big(f(\mathbf{X}, t)\big),$$

and the same holds for the associated probability-flow ODE.

**Conclusion.** Since centering removes translations and commutes with rotations, and all SSD terms are rotation-covariant and translation-invariant in the centered frame (radial contraction, repulsion, and the equivariant score; isotropic noise), Eq. 17 holds. Thus SSD is SE(3)-equivariant on $\mathbb{R}^{n \times 3}/\text{Transl}$, which is the natural domain for molecular conformations and for all evaluations (performed after centering/alignment).

## I    THIN-SHELL FORWARD–REVERSE CONSISTENCY

This appendix provides a proof sketch showing that (i) the SSD forward SDE converges to a thin-shell distribution via a decoupled radial–angular process, and (ii) the reverse SDE used in SSD is a valid time-reversal satisfying marginal consistency under score-based diffusion theory. Together, these results justify the use of shell-based forward and reverse processes.

**Forward SDE.** For each atom, the forward dynamics are

$$d\mathbf{x}_t = \alpha_t \frac{\mathbf{x}_T - \mathbf{x}_t}{\|\mathbf{x}_T - \mathbf{x}_t\|} dt + \sigma_t\, d\mathbf{w}_t, \qquad \sigma_t \ll 1. \tag{20}$$

Let $r_t = \|\mathbf{x}_t\|$ and $\mathbf{u}_t = \mathbf{x}_t/r_t$ be the radial and angular components. Using standard Itô decomposition (e.g., Hsu (2002)), the process separates into

$$dr_t = \alpha_t(r - r_t)\, dt + \sigma_t\, dB_t, \tag{21}$$

$$d\mathbf{u}_t = \sigma_t\, \Pi_{\mathbf{u}_t} \circ\, dW_t, \tag{22}$$

where $B_t$ is a 1D Brownian motion, $\Pi_{\mathbf{u}_t}$ projects onto the tangent space of the sphere, and $W_t$ is Brownian motion on $\mathbb{R}^3$. Eq. equation 21 is an Ornstein–Uhlenbeck (OU) process with attracting point $r$, so $r_t \to r$ exponentially fast. Eq. equation 22 is Brownian motion on the sphere with vanishing diffusion strength $\sigma_t$.

**Convergence to a thin shell.** Because the radial OU drift scales as $\alpha_t(r - r_t)$ and $\sigma_t \ll 1$, the stationary distribution of $r_t$ concentrates around $r$ with width

$$\mathrm{Var}(r_t) \approx \frac{\sigma_t^2}{2\alpha_t}, \tag{23}$$

yielding a spherical shell of thickness $\tau = O(\sigma_t/\sqrt{\alpha_t})$. The angular component remains approximately uniform on the sphere due to Eq. equation 22. Thus, the terminal distribution of the forward SDE is

$$p_T(\mathbf{x}) \approx \mathrm{Unif}(\mathbb{S}^2(r)) * \mathcal{N}(0, \tau^2 I), \tag{24}$$

i.e., an $\tau$-thin spherical shell centered at radius $r$.

**Reverse SDE via score-based time reversal.** Score-based diffusion (Anderson, 1982; Song et al. (2021)) states that reversing an SDE $d\mathbf{x}_t = f(\mathbf{x}_t, t)\,dt + \sigma_t\,d\mathbf{w}_t$ yields the reverse-time SDE

$$d\mathbf{x}_t = \left(f(\mathbf{x}_t, t) - \sigma_t^2 \nabla_{\mathbf{x}} \log p_t(\mathbf{x}_t)\right) dt + \sigma_t\,d\bar{\mathbf{w}}_t, \tag{25}$$

where $p_t$ are the forward marginals. In SSD, the learned score network $s_\theta(\mathbf{x}_t, t)$ estimates $\nabla \log p_t(\mathbf{x}_t)$. Because the forward marginal $p_t$ is concentrated on a thin shell and remains rotationally symmetric, its score naturally decomposes into radial and tangential components:

$$\nabla \log p_t(\mathbf{x}_t) = \beta_t(r_t) \frac{\mathbf{x}_t}{\|\mathbf{x}_t\|} + \text{tangential corrections.} \tag{26}$$

Substituting this into the reverse SDE yields exactly the SSD reverse dynamics from Eq. equation 10: an inward radial attraction, a learned tangential correction $s_\theta$, and repulsion for numerical and chemical stability.

**Forward–reverse marginal consistency.** Since the forward SDE converges to a thin-shell distribution and the reverse SDE is obtained via the standard score-based time reversal, the two processes form a consistent forward–reverse pair:

$$p_{t\,|\,0}^{\text{forward}} = p_{t\,|\,T}^{\text{reverse}} \quad \text{for all } t, \tag{27}$$

up to the small shell-noise perturbation $\sigma_t$. Thus, SSD performs valid generative sampling on a thin shell rather than in the full ambient space, allowing the reverse process to focus on the direct shell-to-structure pathway.

## J  TRAINING DETAILS

Table 6 summarizes the training protocols used for all backbones and our SSD variants. We strictly follow the official implementations (GeoDiff, SubGDiff, EDM, SemlaFlow), only replacing the Gaussian prior with SSD's spherical-shell prior and applying SSD forward/reverse dynamics. Detailed hyperparameters are given below.

Table 6: Training protocols for conditional (Cond.) and unconditional (Uncond.) backbones. SSD variants inherit the same settings, with only initialization and dynamics replaced by SSD.

| Model | Task | Schedule | Steps $T$ | Batch / LR |
|---|---|---|---|---|
| GeoDiff / SubGDiff (Xu et al., 2022; Zhang et al., 2024) | Cond. | 2M (QM9), 6M (Drugs) | 5000 (QM9), 500 (Drugs) | 64, 32 / $2 \times 10^{-4}$ |
| **SSD-GeoDiff / SSD-SubGDiff** | Cond. | same | same | same |
| EDM (Hoogeboom et al., 2022) | Uncond. | 120K (QM9), 200K (Drugs) | 1000 | 64 / $1 \times 10^{-4}$ (cosine) |
| **SSD-EDM** | Uncond. | same | same | same |
| SemlaFlow (Irwin et al., 2025) | Uncond. (flow) | 120K (QM9), 200K (Drugs) | 100 | 64 / $1 \times 10^{-4}$ |
| **SSD-Flow** | Uncond. (flow) | same | same | same |

**Auto-calibrated SSD settings (single pass, no tuning).** We perform a single one-shot calibration on the training split and *fix* all SSD-specific values thereafter; no per-dataset or per-backbone tuning is conducted. Concretely: (i) *Shell radius $r$*: compute each molecule's mean absolute radius about its centroid and set $r$ to the *mean* across training molecules; (ii) *Minimum separation $d_{\min}$*: set to the *minimum bond length observed in the training set* (computed once per dataset); (iii) *Thin-shell noise $\sigma_t$*: used *only* for diffusion backbones to realize a numerically stable shell of negligible thickness; for Flow Matching we set $\sigma_t=0$. The resulting numeric values used in all experiments appear in Table 7.

**Conditional generation (GeoDiff / SubGDiff backbones).** We adopt the official implementations of GeoDiff (Xu et al., 2022) and SubGDiff (Zhang et al., 2024) on QM9 and Drugs, following their preprocessing and splits. Training protocols (iterations, optimizer, warmup, batch sizes, and linear noise schedules) remain unchanged. **SSD-GeoDiff** and **SSD-SubGDiff** replace only the Gaussian prior/dynamics with SSD, using the auto-calibrated values in Table 7.

**Unconditional refinement (EDM backbone).** For unconditional refinement we follow EDM (Hoogeboom et al., 2022) with an EGNN backbone, including optimizer, schedules, and diffusion steps. **SSD-EDM** keeps all settings identical except for SSD spherical-shell initialization and dynamics with the auto-calibrated values (Table 7).

**Unconditional refinement (SSD-Flow).** To extend SSD to Flow Matching, we adopt the objective of Lipman et al. (2023); Irwin et al. (2025) while retaining EDM-style coordinate updates. The SSD drift (Eq. 10) is interpreted as a deterministic velocity field (here $\sigma_t = 0$), and training minimizes the flow-matching loss between shell-initialized samples and ground-truth conformations. We borrow SemlaFlow's substructure-aware encodings (e.g., subgraph pooling and attention) but keep EGNN-based coordinate updates to preserve SE(3)-equivariance. Schedules match SemlaFlow; initialization/dynamics use the same auto-calibration (Table 7).

**Integration with SSD dynamics.** Across all settings, Gaussian-based forward and reverse processes are replaced by SSD dynamics (Eq. 8, Eq. 10). Backbones (GIN (Xu et al., 2019) in GeoDiff/SubGDiff, EGNN (Satorras et al., 2021) in EDM/SSD-Flow) parameterize the score/velocity field that enters the reverse drift. In score-based diffusion, SSD modifies initialization and drift; in Flow Matching, SSD provides structured priors and radial contraction while the learned flow replaces stochastic drift.

**Inference.** Sampling steps are fixed per backbone: $T = 5000$ (SSD-GeoDiff/SSD-SubGDiff), $T = 1000$ (SSD-EDM), and $T = 100$ (SSD-Flow). We use ODE or SDE samplers accordingly; robustness to fewer steps is evaluated in Section 4.4.

**Hardware and environment.** All experiments are run on NVIDIA A100 40GB GPUs with PyTorch 2.1 and CUDA 12, using mixed-precision training.

Table 7: Auto-calibrated SSD values used throughout (distances in Å). Values are computed once on the training split and fixed for all runs. *Note:* $\sigma_t$ applies only to diffusion backbones; for Flow Matching we set $\sigma_t = 0$.

| Dataset | $r$ | $\sigma_t$ | $d_{\min}$ |
|---------|-------|------------|------------|
| QM9 | 6.37 | $10^{-6}$ | 1.07 |
| Drugs | 21.16 | $10^{-6}$ | 1.07 |

## K  ADDITIONAL EVALUATION ON THE CONFORMER FIELDS (MCF) BACKBONE

To assess robustness across evaluation protocols, we additionally apply SSD to the Conformer Fields (MCF) model (Wang et al., 2024) under its *original* GEOM-QM9 evaluation protocol. This protocol uses the same dataset split, the same set of 1000 evaluation molecules, and the same official COV/MAT metrics as reported in the MCF paper.

Table 8 reports results using the standard DDPM sampler with 1000 sampling steps. SSD-MCF improves all four GEOM metrics over the original MCF backbone. Since the MCF paper also discusses DDIM as an alternative sampler, Table 9 evaluates the same backbone with DDIM-50 under identical settings. SSD-MCF again achieves consistent improvements.

These results demonstrate that SSD provides a stable, plug-and-play enhancement to the MCF backbone under *its own native evaluation protocol*, independent of the SubGDiff setting used in the main experiments.

Table 8: Evaluation under the original MCF protocol (GEOM-QM9) using the standard DDPM sampler (1000 steps).

| Method | COV-R ↑ | COV-P ↑ | MAT-R ↓ | MAT-P ↓ |
|---|---|---|---|---|
| Conformer Fields (MCF) (Wang et al., 2024) | 95.00 | 93.70 | 0.1030 | 0.1190 |
| **SSD-MCF (ours)** | **96.15** | **94.61** | **0.0993** | **0.1087** |

Table 9: Evaluation under the original MCF protocol (GEOM-QM9) using the DDIM sampler (50 steps).

| Method | COV-R ↑ | COV-P ↑ | MAT-R ↓ | MAT-P ↓ |
|---|---|---|---|---|
| Conformer Fields (MCF) (Wang et al., 2024) | 92.71 | 92.64 | 0.1168 | 0.1206 |
| **SSD-MCF (ours)** | **93.39** | **92.80** | **0.1146** | **0.1166** |

## L ADDITIONAL ABLATIONS ON SHELL GEOMETRY, DYNAMICS, AND NOISE

This appendix provides extended ablation studies complementing those presented in the main text. We investigate (i) the choice of shell radius for initialization, (ii) the contribution of geometric priors in the reverse process, (iii) sensitivity to the thin-shell noise scale $\sigma_t$, (iv) the effect of atom–shell assignment strategies, and (v) whether SSD's gains can be reproduced by reducing or rescaling Gaussian noise. Unless otherwise stated, all studies use SSD-SubGDiff on GEOM-QM9 following the SubGDiff evaluation protocol.

**Shell radius.** We evaluate three families of radii: a *small shell* (0.01–5Å), a *mean shell* derived from the mean radius of centered training conformations (6.37Å, with a nearby 7Å variant), and a *large shell* (10Å). Table 10 shows that extremely small radii cause atomic collapse and poor coverage, while excessively large radii significantly degrade sample quality. The dataset-adaptive mean shell provides the best balance and is consistently competitive across backbones.

Table 10: Ablation on shell radius for SSD-SubGDiff (QM9). COV-R (higher is better) and MAT-R (lower is better).

| Initialization | COV-R ↑ | MAT-R ↓ |
|---|---|---|
| SubGDiff (Gaussian init) | 90.9 | 0.246 |
| Mean shell (no SSD dynamics) | 91.0 | 0.245 |
| Small shell (0.01 Å) | 75.2 | 0.352 |
| Small shell (1 Å) | 82.0 | 0.310 |
| Small shell (5 Å) | 85.5 | 0.288 |
| Mean shell (6.37 Å) | **93.2** | **0.238** |
| Mean shell (7 Å) | 91.5 | 0.241 |
| Large shell (10 Å) | 84.0 | 0.290 |

**Geometric priors in the reverse dynamics.** With the mean shell fixed, Table 11 evaluates the contribution of each geometric prior: minimum-separation repulsion $d_{\min}$, thin-shell noise $\sigma_t$, and their combination. Repulsion stabilizes close-contact regions and prevents collapse, while thin-shell noise improves coverage by diversifying radial configurations. The combination yields the strongest results, corresponding to the full SSD dynamics.

**Sensitivity to thin-shell noise $\sigma_t$.** We sweep $\sigma_t$ across several orders of magnitude (from $10^{-6}$ to $10^{-1}$), with $\sigma_t = 0$ corresponding to a deterministic shell. Table 12 shows that SSD is robust to $\sigma_t$: even large deviations cause only minor fluctuations, and the default $10^{-6}$ offers the best trade-off. This indicates that SSD's benefits originate from shell-aware geometry rather than noise tuning.

Table 11: Ablation on geometric priors for SSD-SubGDiff (QM9). Mean shell (6.37Å) used for all variants.

| Variant | COV-R ↑ | MAT-R ↓ |
|---|---|---|
| SubGDiff (Gaussian init) | 90.9 | 0.246 |
| Mean shell (no SSD dynamics) | 91.0 | 0.245 |
| Mean shell (no priors) | 91.0 | 0.245 |
| Mean shell + $d_{\min}$ | 93.1 | 0.240 |
| Mean shell + $\sigma_t$ | 92.7 | 0.240 |
| Mean shell + $d_{\min}$ + $\sigma_t$ (full SSD) | **93.2** | **0.238** |

Table 12: Ablation on thin-shell noise scale $\sigma_t$ for SSD-SubGDiff (QM9).

| Variant | COV-R ↑ | MAT-R ↓ |
|---|---|---|
| Mean shell + $d_{\min}$ ($\sigma_t = 0$) | 93.1 | 0.240 |
| Mean shell + $d_{\min}$ + $\sigma_t$ ($10^{-6}$) | **93.2** | **0.238** |
| Mean shell + $d_{\min}$ + $\sigma_t$ ($10^{-5}$) | 93.0 | 0.239 |
| Mean shell + $d_{\min}$ + $\sigma_t$ ($10^{-4}$) | 92.9 | 0.239 |
| Mean shell + $d_{\min}$ + $\sigma_t$ ($10^{-3}$) | 92.8 | 0.240 |
| Mean shell + $d_{\min}$ + $\sigma_t$ ($10^{-2}$) | 92.7 | 0.241 |
| Mean shell + $d_{\min}$ + $\sigma_t$ ($10^{-1}$) | 92.5 | 0.242 |

**Atom–shell assignment strategies.** Because shell points have no semantic ordering, SSD assigns them using a random permutation $\pi$ to preserve permutation symmetry. Table 13 compares this with a deterministic nearest-distance matching (NDM). NDM enforces fixed spatial correspondences between atoms and shell points—a condition that never occurs during inference—which severely distorts the forward geometry and drives trajectories off the shell. As a result, NDM collapses coverage and leads to a catastrophic increase in MAT-R, demonstrating a significant deterioration in matching quality. This outcome confirms that random permutation is critical for maintaining stable and accurate shell dynamics.

Table 13: (Supplemented from rebuttal Table R5) Ablation on atom–shell assignment (GEOM-QM9, SSD-SubGDiff). Random assignment is used in SSD; NDM significantly degrades performance.

| Assignment Strategy | COV-R ↑ | MAT-R ↓ |
|---|---|---|
| Random (our default) | **93.2** | **0.2380** |
| Nearest-distance matching | 0.0 | 1.7470 |

**Reducing Gaussian noise cannot reproduce SSD.** A natural question is whether SSD's benefits arise simply from shrinking the Gaussian noise scale. Table 14 shows that reducing $\sigma_t$ over a wide range ($0.1\times$–$0.9\times$) does not recover SSD's performance: coverage drops substantially, and even the best reduced-noise Gaussian variants remain far below SSD. This confirms that SSD's advantage arises from its geometry-aligned spherical shell, not from variance tuning.

**Summary.** Across all experiments, we observe four consistent principles: (1) radii must match molecular scale; (2) geometric priors such as $d_{\min}$ and $\sigma_t$ stabilize dynamics but are effective only when paired with a shell-aligned forward process; (3) thin-shell noise is not a tuning parameter but a minimal stochastic support for the shell geometry; and (4) SSD's gains cannot be reproduced by simply reducing Gaussian variance or by using deterministic assignment. Overall, these results

Table 14: (Supplemented from rebuttal Table R6) Ablation on Gaussian noise reduction (GEOM-QM9). Shrinking $\sigma_t$ from $0.1\times$ to $0.9\times$ of the original schedule degrades performance and does not match SSD.

| Variant | COV-R ↑ | MAT-R ↓ |
|---|---|---|
| Gaussian baseline | 90.9 | 0.2460 |
| Gaussian ($0.1 \times \sigma_t$) | 60.3 | 0.3755 |
| Gaussian ($0.5 \times \sigma_t$) | 72.4 | 0.3520 |
| Gaussian ($0.7 \times \sigma_t$) | 75.8 | 0.3205 |
| Gaussian ($0.8 \times \sigma_t$) | 78.1 | 0.2720 |
| Gaussian ($0.9 \times \sigma_t$) | 85.3 | 0.2450 |
| **SSD (ours)** | **93.2** | **0.2380** |

show that SSD's improvements come from a geometry-aligned forward–reverse process built on a chemically calibrated spherical shell.

## M  EFFICIENCY ANALYSIS

SSD introduces no additional parameters for coordinate-based diffusion backbones (SubGDiff), leaving both training and inference time unchanged. For latent flow-matching backbones, SSD-Flow replaces SemlaFlow's original encoder with a more compact EDM-style encoder, reducing both parameter count and wall-clock time while preserving the same flow-matching formulation. All SSD variants use the same batch sizes, sampling schedules, and optimizers as their corresponding baselines, and the geometric drift terms reuse pairwise distances already computed by the backbone, adding negligible overhead.

**Training-time and Parameter Comparison.**  Table 15 (supplemented from rebuttal) summarizes model size and wall-clock training time per epoch. SSD-SubGDiff matches SubGDiff exactly in both parameter count and runtime. In contrast, SSD-Flow achieves significantly lower parameter count and faster training time than SemlaFlow due to the more compact EDM-style encoder. Overall, SSD maintains the same computational cost for coordinate models and improves efficiency for latent-flow models, while simultaneously improving geometric stability and accuracy.

Table 15: (Supplemented from rebuttal Table R3) Parameter and runtime comparison across coordinate and latent-flow backbones (GEOM-QM9). SSD preserves coordinate runtime and reduces latent-flow cost.

| Method | Encoder Type | Params (M) | Training Time/epoch (min) | Inference Time/step (ms) |
|---|---|---|---|---|
| SubGDiff | SubGDiff encoder | 0.79 | 2.16 | 0.38 |
| **SSD-SubGDiff** | SubGDiff encoder | **0.79** | **2.16** | **0.38** |
| SemlaFlow | SemlaFlow encoder | 46.60 | 5.01 | 1.03 |
| **SSD-Flow** | EDM-style encoder | **28.54** | **3.79** | **0.84** |

**Inference Cost.**  Sampling cost remains unchanged: SSD uses the same number of SDE/ODE steps as the corresponding backbones, and the analytic radial and repulsion drifts incur negligible overhead compared to the SE(3)-equivariant layers that dominate inference time.

**Summary.**  Across all backbones and datasets, SSD preserves computational cost for coordinate diffusions and improves efficiency for latent-flow models, while simultaneously improving sam-

pling stability and final accuracy. This confirms that SSD's benefits arise from its geometry-aligned formulation rather than increased model capacity or runtime.

## N Trajectory Stability and Spatial Entropy

To provide a unified trajectory-stability analysis across both SDE- and ODE-based samplers, we replace drift-magnitude statistics (which are zero for deterministic ODE flows) with two geometry-aware measures of spatial entropy:

- **Radial deviation:** $\mathbb{E}_t[|\|x_t\| - r_0|]$, which quantifies how far trajectories stray from the target shell geometry.
- **Path excess ratio:** $\mathrm{PathLen}/\|x_T - x_0\|$, which measures the amount of unnecessary wandering relative to the direct path toward the final structure.

Both metrics apply to SDE (GeoDiff) and ODE (SemlaFlow/SSD-Flow) samplers.

**Results.** Across all settings, SSD exhibits substantially smaller radial deviation and shorter excess path length, indicating more concentrated and geometry-aligned trajectories. Gaussian-based priors show large deviations from the chemical radius and long, meandering paths, consistent with their high spatial entropy. Full numerical results are reported in Table 16.

Table 16: (Supplemented from rebuttal Table R7) Trajectory-stability metrics. Lower is better. SSD reduces radial deviation and unnecessary path length in both SDE and ODE regimes (GEOM-QM9).

| Method | Regime | Radial deviation ↓ | Path excess ratio ↓ |
|---|---|---|---|
| SubGDiff | SDE | 0.4534 | 246.91 |
| **SSD-SubGDiff** | SDE | **0.1781** | **1.71** |
| SemlaFlow | ODE | 0.2145 | 1.29 |
| **SSD-Flow** | ODE | **0.1096** | **1.11** |

**Interpretation.** Gaussian-based methods wander far from the chemically meaningful region before converging, whereas SSD remains close to the spherical shell and moves directly toward the molecular conformation. This confirms that SSD suppresses long-range Gaussian drift and provides more stable, low-entropy trajectories.

## O Use of AI-Assisted Writing Tools

In preparing this manuscript, we utilized ChatGPT (OpenAI's language model) as a writing assistant. The tool was specifically employed to refine the grammar, adjust the tone, and improve the overall readability of the paper. Its role was limited to linguistic polishing and enhancing clarity of expression for readers. Importantly, ChatGPT was not involved in generating scientific content, designing experiments, analyzing data, or drawing conclusions. All core ideas, methodology, and results presented in this work are solely the contributions of the authors. The AI tool functioned only as a supplementary aid to ensure smoother academic writing and more effective communication of our findings.

