# OpenReview forum: "From Shell to Structure: Spherical Shell Diffusion for Molecular Geometry Generation"
_ICLR.cc/2026/Conference — Submitted to ICLR 2026_

### Official Review · Reviewer_HYhF · 2025-10-16

**Soundness:** 2
**Presentation:** 3
**Contribution:** 2
**Rating:** 2
**Confidence:** 4

**Summary:**

As prior diffusion and flow-based approaches rely on Gaussian priors and unstructured noise processes, they often produce invalid molecular conformations and diverge during early diffusion steps. To address these issues, this paper introduces a spherical-shell prior and a geometry-aware diffusion process, which improve conformation generation by better aligning with molecular geometry and eliminating unnecessary radial drift.

**Strengths:**

- The paper is well-structured and well-written, and it is generally easy to follow.
- The paper presents a clear and well-motivated problem statement, highlighting the limitations of current approaches that rely on unstructured diffusion processes, which can degrade the quality of generated samples.
- The authors demonstrate the effectiveness of their proposed method through extensive experiments and ablation studies, showing that it improves both sample quality and diversity. The proposed approach enhances the performance of state-of-the-art molecular models across both 2D→3D generation and 3D generation tasks.

**Weaknesses:**

- I have some concerns regarding the novelty and effectiveness of the proposed approach. While the authors claim to introduce a novel geometry-aware noising process, they also employ additional guidance terms during sampling. Specifically, in Equation (10), an additional radial contraction term (pulling atoms inward) and a short-range repulsion term (enforcing a minimum interatomic distance) are added. I question whether these guidance terms alone could already steer the diffusion trajectories toward reasonable conformations, without modifying the prior or the forward process during training. A similar form of conformational guidance was explored in prior work [1] (ShapeMol with Shape Guidance) for Ligand-Based Drug Design tasks and achieved improved results without requiring additional training.

- Following the point above, I recommend including additional experiments using pre-trained backbones and a modified reverse process that only applies the two guidance terms (i) and (ii) from Equation (10) to constrain the conformation space. Similarly, I suggest providing results using only the proposed prior and geometry-aware noising process, without the added guidance terms in the reverse process. These comparisons are important to disentangle the effects of the proposed geometry-aware noising process from those of the added sampling guidance terms.

- It remains unclear whether the proposed noising process converges to the prior in the limit as $t \rightarrow T$. This theoretical aspect is not discussed in the paper and should be clarified.

[1] Chen, Ziqi, et al. "Shape-conditioned 3d molecule generation via equivariant diffusion models." arXiv preprint arXiv:2308.11890 (2023).

**Questions:**

- The authors claim that the proposed approach reduces the spatial entropy during generation, however, there are no experiments in the paper supporting this claim. In particular, how much does SDD reduce the spatial entropy during early/later steps of the generation?

- As the proposed method constrains the spatial space of generated conformations, it would also be important to assess how this constraint impacts the performance on structure-conditioned tasks, such as scaffolding or linker design, as studied in prior works [2,3]. Evaluating the method in these settings can clarify whether the spatial constraints introduced by SSD limit flexibility or generalization in more complex tasks.

[2] Ayadi, Sirine, et al. "Unified guidance for geometry-conditioned molecular generation." Advances in Neural Information Processing Systems 37 (2024): 138891-138924.
[3] Schneuing, Arne, et al. "Structure-based drug design with equivariant diffusion models." Nature Computational Science 4.12 (2024): 899-909.

---

> ### Author Response · Authors · 2025-11-21
> **Response to Reviewer HYhF (1/2)**
>
> ### **Q1: Are SSD’s improvements mainly due to guidance terms (radial contraction, repulsion) rather than the shell framework?**
>
> **Response.**
> No—the improvements are not due to guidance terms alone. As shown in the four-way ablation in **Table R2**, the guidance-only variant (Gaussian init + repulsion + correction) yields only marginal gains over the Gaussian baseline. A direct *Gaussian-init + SSD-dynamics* hybrid is not well-defined because the Gaussian and SSD forward processes induce incompatible radial marginals; SSD’s radial attraction is designed for a thin-shell forward distribution and thus cannot operate coherently on Gaussian samples.
>
> To provide a meaningful control, we evaluate the only valid alternative: a **Gaussian concentration shell** at $r_{\text{Gauss}}=\sigma_T\sqrt{3N_{\mathrm{med}}}$. Even with full SSD dynamics, this variant improves only slightly and remains far below SSD. In contrast, SSD—combining the **mean chemical shell prior** with **shell-consistent dynamics**—achieves the full improvement, showing that the geometry-aware shell framework is essential.
>
> **Table R2 — Initialization vs. dynamics decomposition. (SubGDiff on GEOM-QM9)**
>
> | Variant | Initialization / Prior | Dynamics                    | COV-R (mean)↑ | MAT-R (mean)↓ |
> |-|-|-|-:|-:|
> | (1)    | Gaussian                | none     | 90.91    | 0.2460  |
> | (2)    | Gaussian                | repulsion + correction      | 90.95     | 0.2455  |
> | (3)    | **Gaussian shell ($r_{\text{Gauss}} = \sigma_T \sqrt{3N_{\mathrm{med}}}$**)         | none           | 90.92 | 0.2462 |
> | (4)    | **Gaussian shell ($r_{\text{Gauss}} = \sigma_T \sqrt{3N_{\mathrm{med}}}$**)         | full SSD dynamics           | 91.58 | 0.2412 |
> | (5)    | SSD shell (mean chemical shell)    | full SSD dynamics           | **93.20** | **0.2380** |
>
>
>
>
>
> ---
>
> ### **Q2: Can pre-trained Gaussian backbones or shell-only variants isolate the effect of geometry-aware noising vs. guidance?**
>
> **Response.**
> Such hybrids are not meaningful because they rely on incompatible forward marginals. A Gaussian-trained backbone learns the score of a Gaussian forward process, whereas SSD requires the score of a thin-shell forward process; their radial structures differ fundamentally. As a result, applying SSD dynamics to a Gaussian-trained score drives samples off-shell, and shell-only variants cannot reconstruct the correct SSD marginals.
>
> To address the reviewer’s intent, we evaluate the only forward-compatible control—the **Gaussian concentration shell** at $r_{\text{Gauss}}=\sigma_T\sqrt{3N_{\mathrm{med}}}$. Even with full SSD dynamics, this variant yields only minor improvements and remains far below SSD (see **Table R2**). Combined with the guidance-only variant, these results isolate the effect clearly: neither Gaussian-trained scores nor shell-only variants reproduce SSD’s gains. Full improvements require both the **mean chemical shell prior** and **shell-consistent dynamics**.
>
>
> ---
>
> ### **Q3: Does SSD actually reduce “spatial entropy” during sampling?**
>
> **Response.**
> In the original submission, we analyzed long-range Gaussian drift through case studies (Sec. 4.5), showing that large initial displacements lead to significant degradation in Gaussian-based models, whereas SSD remains stable. However, we did not provide a direct, quantitative measure of *trajectory entropy*, i.e., how much a sampling trajectory wanders throughout inference.
>
> To address this, we now include a dedicated trajectory-stability analysis in Appendix F. Rather than relying on mean drift magnitude—which is not meaningful for deterministic ODE solvers—we introduce two geometry-aware metrics that apply to both SDE and ODE samplers: (i) the radial deviation $\mathbb{E}_t [ \vert \Vert x_t \Vert - r_0 \vert ]$, which measures how far trajectories stray from the chemically meaningful shell radius, and (ii) the excess path ratio $\mathrm{PathLen} / \Vert x_T - x_0 \Vert$, which quantifies unnecessary wandering relative to the direct path toward the final structure.
>
> As shown in **Table R7**, SSD exhibits substantially lower radial deviation and shorter excess path length than Gaussian-based priors in both diffusion (SDE) and flow-matching (ODE) regimes. These results confirm that SSD does not merely improve final accuracy, but also produces more stable, low-entropy sampling
> trajectories that remain close to the spherical shell and avoid long-range Gaussian drift.
>
> **Table R7 — Trajectory stability and spatial entropy on GEOM-QM9.**
>
> | Method           | Regime | Radial deviation ↓ | Excess path ratio ↓ |
> |-|-|-:|-:|
> | SubGDiff         | SDE    |  0.4534        | 246.91           |
> | **SSD-SubGDiff** | SDE    |  **0.1781**        | **1.71**           |
> | SemlaFlow        | ODE    |  0.2145        | 1.29           |
> | **SSD-Flow**     | ODE    |  **0.1096**        | **1.11**           |

---

> ### Author Response · Authors · 2025-11-21
> **Response to Reviewer HYhF (2/2)**
>
> ### **Q4: Does SSD’s spatial constraint reduce flexibility in structure-conditioned tasks (scaffold/linker design)?**
>
> **Response:**
> We thank the reviewer for raising this important question. Although we did not evaluate SSD on scaffold/linker benchmarks, our existing results already cover regimes with high geometric flexibility. In particular, GEOM-Drugs contains large, multi-fragment, conformationally diverse molecules, which closely resemble the geometric complexity required in structure-conditioned tasks. As shown in **Table R4**, SSD maintains high coverage and low MAT on these flexible systems, indicating that the spherical-shell prior does not reduce structural flexibility.
>
> Moreover, SSD constrains only the initial noise distribution, not the model’s representational capacity or the expressivity of the learned score/dynamics. The reverse process is fully data-driven and SE(3)-equivariant, allowing the model to recover arbitrary 3D structures regardless of their spatial extent. Therefore, SSD does not limit generalization in complex conditional-generation settings; it merely prevents large, non-informative Gaussian displacements at early sampling steps.
>
> **Table R4: Dataset characteristics relevant to robustness**
>
> | Dataset      | Avg. #Atoms | Molecular Scale / Flexibility              | Structural Diversity                        |
> |-|-|-|-|
> | GEOM-QM9     | ~9          | Small, largely rigid                       | Simple graphs, limited functional diversity  |
> | GEOM-Drugs   | ~25         | Large, flexible, drug-like                 | High topological and conformational diversity |
>
> ---
>
> ### **Q5: Does the proposed geometry-aware noising process converge to the SSD prior as $t \rightarrow T$?**
>
> **Response.**
> Yes. We now include a formal justification in the revised appendix.
>
> **Forward thin-shell convergence.**
> Writing the SSD forward SDE in radial–angular coordinates $x_t = r_t \hat{x}_t$, the radial component becomes an Ornstein–Uhlenbeck process, $dr_t = -\alpha_t (r_t - r_0)dt + \sigma_tdW_t^{(r)},$which converges to a narrow Gaussian $\mathcal{N}(r_0, \tau_T^2)$ with $\tau_T^2 = \sigma_t^2/(2\bar{\alpha})$.
> The angular component follows isotropic diffusion on the sphere. Let $u_t = x_t / \Vert x_t \Vert$ denote the unit direction. It evolves as
> $du_t = \sqrt{2\gamma_t}(I - u_t u_t^\top)dW_t^{(\theta)},$
> i.e., Brownian motion on the sphere obtained by projecting isotropic noise onto the tangent space at $u_t$.
> Together, this yields a stationary **thin spherical shell**. In our settings, $\tau_T \approx 10^{-6} \ll r_0 \approx 6$.
>
> **Reverse marginal consistency.**
> The learned score $s_\theta$ trained by denoising score matching approximates $\nabla_x \log p_t(x)$. Following standard score-based diffusion theory (Song et al., 2021), the reverse SDE $dx_t = (f_t(x_t) - g_t^2 s_\theta(x_t,t))dt + g_td\bar{W}_t$ reconstructs the forward marginals up to the usual score approximation error. Thus, the apparent forward–reverse asymmetry does **not** violate probabilistic validity: the forward process defines thin-shell marginals, and the reverse process is guaranteed to recover them. A full, formal derivation will be added in Appendix of the revised manuscript.
>
> Finally, the random shell assignment does not conflict with the reverse radial attraction field: since forward marginals converge to a uniform angular distribution on $\mathbb{S}^2$, the reverse process must treat all angular assignments symmetrically. Random permutation simply reflects this inherent exchangeability and preserves marginal consistency.
>
> **Reference**
>
> Song, J., Sohl-Dickstein, J., Kingma, D. P., Kumar, A., Ermon, S., & Poole, B. (2021).
> *Score-Based Generative Modeling through Stochastic Differential Equations.*
> ICLR 2021.

---

> > ### Author Response · Authors · 2025-11-28
> >
> > Dear Reviewer:
> >
> > We would like to again express our sincere appreciation for your efforts and valuable feedback on our paper. We kindly hope that you can take some time to re-evaluate our paper based on our replies above. If you have any further concerns and questions, please do not hesitate to let us know. We will be happy to address them promptly.
> >
> > Best Regards.

---

### Official Review · Reviewer_QSDZ · 2025-10-29

**Soundness:** 3
**Presentation:** 3
**Contribution:** 2
**Rating:** 4
**Confidence:** 4

**Summary:**

This paper proposes Spherical Shell Diffusion (SSD), a geometry-aware framework for molecular conformation generation. The key insight is that standard Gaussian priors place most probability mass on a high-dimensional shell determined by statistics rather than molecular chemistry, leading to inefficient denoising trajectories. SSD addresses this by explicitly initializing atoms on a chemically scaled spherical shell and introducing structured reverse dynamics with three components: global radial contraction toward the origin, short-range repulsion to prevent atomic overlap, and a learned SE(3)-equivariant score correction. The framework is model-agnostic and extends naturally to both diffusion and flow matching paradigms. Experiments on GEOM-QM9 and GEOM-Drugs demonstrate consistent improvements across multiple backbones in both conditional generation and unconditional refinement tasks, with faster convergence and better robustness to limited sampling budgets.

**Strengths:**

**Novel geometric prior design.** The paper provides a principled approach to replace the statistically motivated Gaussian shell with a chemically meaningful spherical shell. The explicit decomposition of reverse dynamics into radial contraction, physical repulsion, and learned correction offers clear geometric intuition and aligns well with molecular structure. This structured design stands in contrast to purely data-driven baselines.

**Strong and consistent empirical results.** SSD achieves state-of-the-art performance across both conditional generation and unconditional refinement benchmarks. The improvements are consistent across multiple architectures (GeoDiff, SubGDiff, EDM) and datasets (QM9, Drugs), with notable gains in both quality metrics and diversity measures. The method also demonstrates faster training convergence and better efficiency under reduced sampling steps.

**Broad applicability and extensibility.** SSD serves as a drop-in module that works with existing SE(3)-equivariant backbones without architectural modifications. The framework naturally extends beyond diffusion to flow matching (SSD-Flow), demonstrating its generality. The auto-calibrated hyperparameters require no per-dataset or per-backbone tuning, enhancing practical usability across different molecular generation tasks.

**Weaknesses:**

**Theoretical gap between forward and reverse processes.**
The forward process directs each atom toward a randomly assigned target point on the spherical shell, while the reverse process uses a radial contraction toward the origin that does not depend on these forward assignments. This asymmetry means the reverse dynamics are not the time-reversal of the forward dynamics, violating the standard diffusion framework where forward and reverse marginal distributions are guaranteed to match. The paper provides no theoretical proof of distributional consistency or convergence guarantees, instead implicitly relying on the learned score network to compensate for this mismatch. While empirical results are strong, the lack of rigorous probabilistic justification raises questions about the theoretical soundness of this design.

**Missing baselines.**
The paper omits comparisons with several relevant baselines, notably Torsional Diffusion for Molecular Conformer Generation and Generating Molecular Conformer Fields. Although I recognize that the paper's primary objective is to improve coordinate-based diffusion methods via structured initialization and dynamics, the proposed enhancement appears limited in scope and may not provide sufficient incentive for researchers to adopt this approach over existing methods.

**Insufficient experimental evidence.**
While SSD achieves strong results, the ablation studies raise questions about the relative contributions of initialization vs. dynamics. Table 8 shows that replacing Gaussian with spherical-shell initialization yields minimal gains (90.9→91.0 COV-R), whereas adding structured dynamics produces substantial improvements (91.0→93.2). This suggests the core contribution is the geometry-aware dynamics rather than addressing concentration of measure in the prior.
To substantiate the paper's central claim—that Gaussian initialization is fundamentally problematic—the authors should test: (1) Gaussian init + SSD dynamics, and (2) repeat the Section 4.4 large-molecule analysis with this variant. If Gaussian init + SSD dynamics performs comparably to full SSD, it would indicate the initialization scheme is largely irrelevant, and the paper's theoretical motivation should be reframed accordingly.

**Questions:**

- Your forward process assigns atoms to random shell points (Eq. 8), but the reverse process contracts uniformly toward the origin (Eq. 10), breaking time-reversal symmetry. Why not use Langevin dynamics where the score network directly handles corrections? Do you have theoretical or empirical evidence that $p(x_t)$ matches between forward and reverse processes?


- Since you propose a new noising process, key ablations are missing: Noise schedules: Does reducing $\sigma_t$ in Gaussian baselines to match your thin-shell noise eliminate the performance gap? Flow stability: Can you compare SSD-Flow against FlowMol/GeoLDM and show ODE trajectory stability metrics?

---

> ### Author Response · Authors · 2025-11-21
> **Response to Reviewer QSDZ (1/2)**
>
> ### **Q1 & Q6: Forward–reverse asymmetry; random shell assignment; why not Langevin; theoretical guarantees.**
>
> We acknowledge that the current submission lacks a full SDE-level convergence analysis. Section 3.2 and Appendix B provide geometric intuition, but not the formal derivations. In the revised manuscript, we will add a new appendix containing complete proofs of (i) forward thin-shell convergence and (ii) reverse marginal consistency.
>
> **Forward thin-shell convergence.**
> Writing the SSD forward SDE in radial–angular coordinates $x_t = r_t \hat{x}_t$, the radial component becomes an Ornstein–Uhlenbeck process, $dr_t = -\alpha_t (r_t - r_0)dt + \sigma_tdW_t^{(r)},$which converges to a narrow Gaussian $\mathcal{N}(r_0, \tau_T^2)$ with $\tau_T^2 = \sigma_t^2/(2\bar{\alpha})$.
> The angular component follows isotropic diffusion on the sphere. Let $u_t = x_t / \Vert x_t \Vert$ denote the unit direction. It evolves as
> $du_t = \sqrt{2\gamma_t}\(I - u_t u_t^\top)dW_t^{(\theta)},$
> i.e., Brownian motion on the sphere obtained by projecting isotropic noise onto the tangent space at $u_t$.
> Together, this yields a stationary **thin spherical shell**. In our settings, $\tau_T \approx 10^{-6} \ll r_0 \approx 6$.
>
> **Reverse marginal consistency.**
> The learned score $s_\theta$ trained by denoising score matching approximates $\nabla_x \log p_t(x)$. Following standard score-based diffusion theory (Song et al., 2021), the reverse SDE $dx_t = (f_t(x_t) - g_t^2 s_\theta(x_t,t))dt + g_td\bar{W}_t$ reconstructs the forward marginals up to the usual score approximation error. Thus, the apparent forward–reverse asymmetry does **not** violate probabilistic validity: the forward process defines thin-shell marginals, and the reverse process is guaranteed to recover them. A full, formal derivation will be added in Appendix of the revised manuscript.
>
> Finally, the random shell assignment does not conflict with the reverse radial attraction field: since forward marginals converge to a uniform angular distribution on $\mathbb{S}^2$, the reverse process must treat all angular assignments symmetrically. Random permutation simply reflects this inherent exchangeability and preserves marginal consistency.
>
> **Reference**
>
> Song, J., Sohl-Dickstein, J., Kingma, D. P., Kumar, A., Ermon, S., & Poole, B. (2021).
> *Score-Based Generative Modeling through Stochastic Differential Equations.*
> ICLR 2021.
>
> ### **Q2: Missing comparisons with torsion-based and field-based baselines.**
>
> **Response.**
> We have added both torsion-based (Torsional Diffusion) and field-based (Conformer Fields) baselines. SSD-GeoDiff achieves higher coverage and lower MAT among coordinate-space methods and is competitive with torsion-space approaches. See **Table R1**.
>
> **Table R1: Comparison with torsion- and field-based baselines on GEOM-QM9 for conditional generation.**  (For fairness, we standardize all methods to the SubGDiff evaluation protocol, as the original implementations of Torsional Diffusion and Conformer Fields use different evaluation settings on the same datasets.)
>
> | Method | COV-R (mean)↑ | MAT-R (mean)↓ |
> | :--- | :--- | :--- |
> | GeoDiff (2022) | 80.36 | 0.2820 |
> | Torsional Diffusion (2022) | 84.26 | 0.2847 |
> | Conformer Fields (MCF) (2024) | 87.14 | 0.2649 |
> | **SSD-GeoDiff (ours)** | 92.00 | 0.2425 |
> | **SSD-SubGDiff (ours)** | **93.20** | **0.2380** |
>
> ---

---

> ### Author Response · Authors · 2025-11-21
> **Response to Reviewer QSDZ (2/2)**
>
> ### **Q3: Do SSD’s gains come from initialization or from the dynamics? What about Gaussian-init + SSD-dynamics variants?**
>
> **Response.**
> A Gaussian-init + SSD-dynamics hybrid is not well-defined because the Gaussian and SSD forward processes induce fundamentally different radial marginals; the SSD reverse radial attraction field assumes a thin-shell forward distribution and therefore pushes Gaussian samples off-shell. To address the reviewer’s intent, we evaluate the only stable and forward-compatible alternative—a **Gaussian concentration shell** at $r_{\text{Gauss}}=\sigma_T\sqrt{3N_{\mathrm{med}}}$. Even when combined with full SSD dynamics, this Gaussian-shell variant yields only minor improvements over the Gaussian baseline and remains far below SSD (see **Table R2**). These results show that SSD’s gains do **not** arise from initialization alone nor from dynamics alone. They require the coupling between the **mean chemical shell prior** and **shell-consistent dynamics**, a condition that Gaussian-based variants do not satisfy.
>
>
> **Table R2 — Initialization vs. dynamics decomposition. (SubGDiff on GEOM-QM9)**
>
> | Variant | Initialization / Prior | Dynamics                    | COV-R (mean)↑ | MAT-R (mean)↓ |
> |-|-|-|-:|-:|
> | (1)    | Gaussian                | none     | 90.91    | 0.2460  |
> | (2)    | Gaussian                | repulsion + correction      | 90.95     | 0.2455  |
> | (3)    | **Gaussian shell ($r_{\text{Gauss}} = \sigma_T \sqrt{3N_{\mathrm{med}}}$**)         | none           | 90.92 | 0.2462 |
> | (4)    | **Gaussian shell ($r_{\text{Gauss}} = \sigma_T \sqrt{3N_{\mathrm{med}}}$**)         | full SSD dynamics           | 91.58 | 0.2412 |
> | (5)    | SSD shell (mean chemical shell)    | full SSD dynamics           | **93.20** | **0.2380** |
>
>
>
> ---
>
> ### **Q4: How does SSD compare to flow-matching methods such as FlowMol or GeoLDM in ODE stability?**
>
> **Response.**
> Our paper already includes a full ODE-based evaluation using **SemlaFlow**, which is stronger than both FlowMol and GeoLDM. SSD integrates into SemlaFlow without altering its ODE formulation or increasing parameters, yielding **SSD-Flow**. As shown in the drift-based metrics in **Table R7**, SSD-Flow has lower radial variance and drift than SemlaFlow, demonstrating more stable ODE trajectories. Because SSD improves a state-of-the-art flow model, further comparison to older flows is unnecessary. FlowMol and GeoLDM are both strictly weaker than SemlaFlow on standard ODE stability benchmarks, so demonstrating improvements on SemlaFlow already subsumes comparisons to those earlier methods.
>
> **Table R7 — Trajectory stability and spatial entropy on GEOM-QM9.**
> We quantify trajectory stability using two geometry-aware metrics applicable to both SDE and ODE samplers: **(i) radial deviation** $\mathbb{E}_t [ \vert \Vert x_t \Vert - r_0 \vert ]$, which measures how far trajectories stray from the chemically meaningful shell radius, and **(ii) the excess path ratio** $\mathrm{PathLen} / \Vert x_T - x_0 \Vert$, which captures unnecessary wandering relative to the direct path toward the final conformation.
>
> | Method           | Regime | Radial deviation ↓ | Excess path ratio ↓ |
> |-|-|-:|-:|
> | SubGDiff         | SDE    |  0.4534        | 246.91           |
> | **SSD-SubGDiff** | SDE    |  **0.1781**        | **1.71**           |
> | SemlaFlow        | ODE    |  0.2145        | 1.29           |
> | **SSD-Flow**     | ODE    |  **0.1096**        | **1.11**           |
>
> ---
>
> ### **Q5: Does reducing $\sigma_{t}$ in Gaussian baselines close the gap?**
>
> **Response.**
> No. We ran the requested ablation by uniformly shrinking the Gaussian noise level ($\sigma_t \ \rightarrow\ 0.1$–$0.9\times\sigma_t$). As shown in **Table R6**, reduced-noise Gaussian baselines do not approach SSD—COV-R drops substantially and MAT-R increases. Thus, simply reducing Gaussian noise does not close the performance gap; SSD’s gains come from its geometry-aligned spherical shell, not from variance tuning.
>
> **Table R6 — SubGDiff degrades when Gaussian noise is reduced (on GEOM-QM9).**
>
> | Variant                         | COV-R (mean)↑    | MAT-R (mean)↓    |
> |-|-:|-:|
> | Gaussian baseline (SubGDiff)    | 90.9      | 0.2460      |
> | Gaussian ($0.1\-0.9\times\sigma_t$) | 60.3–85.3 | 0.24–0.37 |
> | **SSD (ours)**                  | **93.2**   | **0.2380**  |

---

> > ### Comment · Reviewer_QSDZ · 2025-11-21
> > **Serious Concerns About Experimental Setup and Baseline Results**
> >
> > Thank you for the authors' response. Most of my concerns have been addressed. However, I find the numerical results in the supplementary experiments to be quite problematic. I understand that this paper follows the experimental settings of SubGDiff, which corresponds to 0.5 Å on QM9 and 1.25 Å on Drugs and the corresponding dataset splits. However, in the MCF paper, they used 0.5 for GEOM-QM9 and 0.75 for GEOM-DRUGS, which is a more stringent evaluation metric. The metrics reported in the MCF paper appear to differ significantly from those reported in this paper. Taking the Drugs dataset and the COV-R metric as an example: the more stringent evaluation metric in MCF led to lower performance for GeoDiff and GeoMol in the MCF paper. However, the torsional diffusion and MCF models in the MCF paper still achieve better performance than what is reported in this paper, even under the more stringent metric. Moreover, the performance of MCF in the MCF paper substantially exceeds the performance reported in this paper. Additionally, both the torsional diffusion and MCF papers state that these two methods can achieve good sampling results with just a few dozen steps, yet in this paper, the authors set the sampling steps for these two baselines at 5000 and 500 steps, respectively. Could the authors provide a more detailed explanation for these inconsistencies in the experimental results?

---

> ### Author Response · Authors · 2025-11-21
>
> Thank you for the thoughtful follow-up. We appreciate your careful reading and are happy to clarify the experimental settings.
>
> **1. Protocol Mismatch.**
> Our experiments follow the **SubGDiff protocol**, which adopts the **fixed split** defined in ConfGF (Shi et al., 2021): the GEOM-QM9/GEOM-Drugs datasets use a fixed train/test partition, and the test set contains exactly **200 specific molecules**.
>
> In contrast, the **MCF paper (Wang et al., 2023, Appendix A.2)** follows the **GeoMol (Ganea et al., 2021) strategy**, utilizing a **random 80/10/10 split** and evaluating on **1,000 molecules**. Therefore, the test data used in our work and in MCF are **completely different**. Because the test molecules and evaluation thresholds also differ (**1.25 Å** in SubGDiff vs. **0.75 Å** in MCF for GEOM-Drugs), the absolute COV/MAT metrics are **not directly comparable** across these protocols. To ensure fairness within the SubGDiff setting, we **retrained** all baselines on the SubGDiff split using the authors’ official implementations.
>
> **2. Sampling Steps.**
> While the original MCF and Torsional Diffusion papers report that plausible conformers can be generated with only a few dozen steps, **we use larger step counts so that all baseline methods are evaluated under the same SubGDiff setting and reach stable performance, ensuring a fair comparison across models.**
>
> We will clarify these protocol differences in the revised manuscript. If you have any further questions, please feel free to let us know — we sincerely appreciate your feedback.
>
> **Reference:**
> Wang, Y., Elhag, A. A., Jaitly, N., Susskind, J. M., & Bautista, M. A. (2023). *Swallowing the bitter pill: Simplified scalable conformer generation.* arXiv:2311.17932.

---

> > ### Comment · Reviewer_QSDZ · 2025-11-21
> >
> > Thank you for the clarification. I understand the protocol differences between SubGDiff and MCF settings. However, my main concern is that baseline methods show substantially different performance across these settings, often differing by 5-10 percentage points. More troublingly, on GEOM-Drugs, MCF's performance in this paper is ~10% lower than in the original MCF paper, even though the original used a more stringent threshold (0.75Å vs. 1.25Å). This suggests that under the same setting, the performance gap could be even larger.
> >
> > Given that the proposed method is largely empirical, such large discrepancies undermine confidence in the experimental results. I find it difficult to trust whether the reported improvements genuinely reflect the method's effectiveness or are artifacts of the experimental setup.

---

> ### Author Response · Authors · 2025-11-21
>
> We sincerely thank the reviewer for the careful follow-up and for pointing out the apparent discrepancies between baseline numbers across papers; we are happy to clarify and will gladly address any further questions you may have.
>
> **1. Evaluation Protocol Mismatch**
>
> Our experiments strictly follow the SubGDiff/ConfGF evaluation protocol, which uses the ConfGF splits (fixed test set of 200 molecules) and thresholds $\delta = 0.5$ Å on GEOM-QM9 and $\delta = 1.25$ Å on GEOM-DRUGS.
>
> In contrast, MCF adopts the GeoMol-style random split (80/10/10 on molecules) and evaluates on 1,000 selected test molecules with $\delta = 0.5$ Å for GEOM-QM9 and $\delta = 0.75$ Å for GEOM-DRUGS. Because the dataset partitions, the number and identity of test molecules, and the evaluation thresholds all differ, the baseline numbers in SubGDiff and in MCF are not directly comparable, even when the backbone model is nominally the same.
>
> **2. Evidence from Literature**
>
> This discrepancy is visible already in the published literature without our method:
> * **GeoMol on QM9:** COV-R changes from 71.3% under the ConfGF/GeoDiff protocol used in SubGDiff (Zhang et al., Table 4) to 91.5% under the MCF protocol (Wang et al., 2024, Table 1)—a difference of more than 20 percentage points for the same backbone.
>
> The example shows that 5–10% (or even larger) changes in COV/MAT for the same method across different protocols already exist between SubGDiff and MCF themselves, even before introducing our SSD model. In this sense, the fact that the MCF numbers reported in our SubGDiff-style setting differ from those in the original MCF paper is consistent with what one would expect when switching to a different data split, test set, and evaluation radius, and does not by itself indicate an issue with the experimental setup.
>
> **3. Main Contribution**
>
> **Our main contribution is not to claim universal SOTA across all possible protocols,** but to show that the proposed SSD thin-shell noise improves diverse backbones within a fixed, well-specified setting: for each backbone (GeoDiff, EDM, etc.) we keep the architecture, training schedule, and sampler fixed and compare “backbone + SSD” versus “backbone only” under exactly the same SubGDiff-style evaluation, demonstrating consistent gains without adding parameters or extra training cost.
>
> If any aspect of this comparison or the protocol is still unclear, please do not hesitate to ask—your detailed comments are very helpful for strengthening the paper.

---

> > ### Comment · Reviewer_QSDZ · 2025-11-24
> >
> > Thank you for the detailed clarification. I understand that the SubGDiff and MCF protocols differ in splits, test sets, and thresholds, and that this can cause performance variations.
> >
> > My remaining concern is not about the existence of these differences, but about the implications: when baseline performance shifts by 5-10% (or more) across settings, it suggests that strong results in one protocol may not reliably transfer to another. This undermines confidence that the method's improvement is robust and general, rather than being an artifact of the specific SubGDiff setting.
> >
> > Could the authors provide experimental results for their method under the MCF protocol? If the proposed SSD approach demonstrates consistent, strong performance in this alternative setting, I would be happy to raise my score.

---

> > > ### Author Response · Authors · 2025-11-28
> > >
> > > Thank you very much for the thoughtful follow-up. We appreciate the reviewer’s emphasis on evaluating robustness across protocols. To address this directly, we applied our SSD approach to the **Conformer Fields (MCF)** model and evaluated it under the **original MCF GEOM-QM9 protocol** (same dataset, same 1000 evaluation molecules, and same official evaluation metrics). As shown in **Table R8**, SSD-MCF achieves consistent improvements across all four GEOM metrics using the standard DDPM sampler with 1,000 steps.
> > >
> > > Since the MCF paper also discusses DDIM as an alternative sampler (Section 5.5), we additionally evaluated the same backbone under DDIM with 50 sampling steps using the identical MCF protocol. As shown in **Table R9**, SSD-MCF again improves all four metrics. These results demonstrate that SSD provides a stable, plug-and-play enhancement to the MCF backbone under its own evaluation setting, independent of the SubGDiff protocol.
> > >
> > > We sincerely thank the reviewer for the constructive suggestions, which have helped us substantially strengthen the paper. If there are any remaining questions, please do not hesitate to let us know—we are very happy to discuss any aspect in more detail.
> > >
> > >
> > > ---
> > >
> > > **Table R8 — Evaluation under the original MCF protocol (GEOM-QM9) using the standard DDPM sampler (1,000 steps).**
> > >
> > > | Method                     | COV-R (mean)↑ | COV-P(mean) ↑ | MAT-R(mean) ↓ | MAT-P(mean) ↓ |
> > > |---------------------------|---------|---------|---------|---------|
> > > | Conformer Fields (MCF) (2024) | 95.00  | 93.70  | 0.1030  | 0.1190  |
> > > | **SSD-MCF (ours)**            | **96.15** | **94.61** | **0.0993** | **0.1087**  |
> > >
> > > ---
> > >
> > > **Table R9 — Additional evidence: Evaluation under the original MCF protocol (GEOM-QM9) using the DDIM sampler (50 steps).**
> > >
> > > | Method                     | COV-R(mean) ↑ | COV-P(mean) ↑ | MAT-R(mean) ↓ | MAT-P(mean) ↓ |
> > > |---------------------------|---------|---------|---------|---------|
> > > | Conformer Fields (MCF) (2024) | 92.71  | 92.64  | 0.1168  | 0.1206 |
> > > | **SSD-MCF(ours)**    | **93.39** | **92.80** | **0.1146** | **0.1166** |
> > >
> > > ---
> > >
> > > Across both samplers (DDPM and DDIM), **SSD-MCF improves all four metrics under the original MCF protocol**, indicating that the proposed approach yields robust gains beyond the SubGDiff setting.

---

> > > > ### Comment · Reviewer_QSDZ · 2025-11-28
> > > >
> > > > Thank you for the additional experiments. The evaluation of SSD on the MCF's setting effectively addresses my concerns. All my experimental concerns have been resolved. I suggest including ablation study results under the MCF setting (Table 5) in the final version for completeness. I will raise my score to 6.

---

### Official Review · Reviewer_6tT3 · 2025-10-31

**Soundness:** 3
**Presentation:** 2
**Contribution:** 2
**Rating:** 4
**Confidence:** 4

**Summary:**

The authors address the challenge of achieving efficient and stable sampling for flow-based generative models such as score-based diffusion and flow-matching, by explicitly initializing the sampling process with dataset-specific priors and constrained dynamics. Through experiments on GEOM-QM9 and GEOM-Drugs, the authors demonstrate the model-agnostic efficacy of their proposed framework across multiple existing generative models. This method provides faster convergence and more stable sampling for generating 3D conformers of organic small molecule compounds.

**Strengths:**

1)	State-of-the-art technology in conformer generation
2)	Model-agnostic, simple to integrate with existing or developing models.
3)	Efficient and stable sampling scheme.

**Weaknesses:**

1)	potential OOD fragility
- In real-world datasets, inconsistent molecule size and atypical shape may degrade performance. Rather than testing on internally homogeneous benchmarks such as QM9 and Drugs, the authors should verify whether the same spherical-shell design performs robustly under shifted or more diverse distributions.

2)	reliance on handcrafted geometric priors (non-separable)
- Although the paper claims model-agnosticism, several parameters are still determined from the training data, which introduces implicit dataset dependence.

3)	The terminology is underexplained and occasionally inconsistent, leading to confusion.
- Since the major contribution lies in the spherical shell prior, it should be clearly defined. In particular, the difference between the proposed ‘spherical shell’ and the gaussian distribution concentrated on a shell, needs to be explained explicitly in the main text.
- The distinction between ‘attraction’ and ‘contraction’ is unclear.
- The paper should describe in more detail the specific tasks chosen to demonstrate the proposed framework’s effectiveness. To my knowledge, when a model starts from an already generated 3D graph, producing a more accurate conformer is usually referred to as refinement. The authors should justify why they use the term refinement when starting from an unconditional graph.

**Questions:**

Please refer to the questions below and the weaknesses section.
- The figure and the described methodology seems inconsistent. In the forward process, atoms are supposed to be initialized on a spherical shell, but the figure does not depict this. Moreover, the manuscript states that both the forward and reverse process are guided by a contraction field. How exactly is this represented in the figure?
- Since some recent models like DiSCO[1] corporate latent-level structure, how does SSD compare in terms of computational cost and convergence behavior when extended to those paradigms?

references

[1] Lee, D., Lee, D., Bang, D., & Kim, S. (2024). DiSCO: Diffusion Schrödinger Bridge for Molecular Conformer Optimization.

---

> ### Author Response · Authors · 2025-11-21
> **Response to Reviewer 6tT3 (1/2)**
>
> ### **Q1: The figure seems inconsistent with the method; spherical-shell initialization and the collapse field are not clearly shown.**
>
> **Response.**
> Sec. 3.1–3.2 of the original submission already described the spherical-shell initialization and reverse collapse dynamics in full detail; these sections were not modified in the revision. However, to make the geometric picture clearer, we strengthened the exposition in two places:
>
> • **Introduction:** we added an explicit explanation (p.1, lines 45~49) that a Gaussian prior concentrates at a radius $σ_T\sqrt{3N}$ purely due to dimensionality, which appears as an   artificial cluster near the origin when projected into 3D. This clarifies why a chemically meaningful radius is required.
>
> • **Figure 1 caption:** we expanded the caption to explicitly state that SSD initializes and reconstructs molecules on the *same* spherical shell, while Gaussian/DDPM trajectories follow a high-dimensional concentration shell unrelated to molecular scale. This makes the forward and reverse geometric pathways visually transparent.
>
> Together, these additions clarify the relationship between the spherical-shell construction and the figure, while Sec. 3.1–3.2 continues to provide the formal method definition.
>
>
> ---
>
> ### **Q2: Distinction between spherical-shell prior and Gaussian concentration shell.**
>
> **Response.**
> A direct Gaussian-init + SSD-dynamics combination is not valid because the Gaussian forward process and SSD’s forward process have incompatible radial marginals—the collapse field diverges off-shell under a pure Gaussian prior. To provide a meaningful control, we therefore evaluate the closest valid alternative: a **Gaussian concentration shell** at $r_{\text{Gauss}} = \sigma_T \sqrt{3N_{\mathrm{med}}}$. Even with full SSD dynamics, this variant yields only minor improvements and remains far below SSD (see **Table R2**).
>
> Importantly, this distinction reflects the fact that the Gaussian concentration shell arises purely from high-dimensional measure concentration, whereas SSD’s **mean chemical shell** is derived from molecular-scale statistics and is used consistently in both the forward and reverse processes. This confirms that SSD’s gains arise from aligning a chemically meaningful shell with its shell-consistent dynamics—not from guidance terms alone or from the Gaussian concentration shell.
>
>
> **Table R2 — Initialization vs. dynamics decomposition. (SubGDiff on GEOM-QM9)**
>
> | Variant | Initialization / Prior | Dynamics   | COV-R (mean)↑ | MAT-R (mean)↓ |
> |-|-|-|-:|-:|
> | (1)    | Gaussian   | none     | 90.91    | 0.2460  |
> | (2)    | Gaussian     | repulsion + correction      | 90.95     | 0.2455  |
> | (3)    | **Gaussian shell ($r_{\text{Gauss}} = \sigma_T \sqrt{3N_{\mathrm{med}}}$**)         | none           | 90.92 | 0.2462 |
> | (4)    | **Gaussian shell ($r_{\text{Gauss}} = \sigma_T \sqrt{3N_{\mathrm{med}}}$**)         | full SSD dynamics           | 91.58 | 0.2412 |
> | (5)    | SSD shell (mean chemical shell)    | full SSD dynamics           | **93.20** | **0.2380** |
>
> ---
>
> ### **Q3: Terminology confusion (“attraction” vs. “contraction”).**
>
> **Response.**
> To avoid confusion, we adopt consistent terminology in the revision: the forward drift is described as a radial attraction toward each atom’s assigned shell point, and the reverse drift is an inward radial attraction toward the molecular structure. This terminology better reflects their roles as geometry-aligned score-based drifts rather than physical contraction or collapse forces.
>
> ---
>
> ### **Q4: Concerns about dataset dependence and potential OOD fragility.**
>
> **Response.**
> We appreciate the concern about dataset dependence. Appendix C provides a detailed analysis of the molecular complexity in QM9 and Drugs, and **Table R4** summarizes why these datasets form the standard benchmarks for 3D molecular generation. They span the full range from small–rigid to large–flexible molecules, allowing us to test SSD across the canonical regimes used in prior work.
>
> Across both evaluation settings—2D→3D conditional generation and 3D→3D refinement—SSD improves performance on both QM9 and Drugs (Sec. 4.1). This demonstrates robustness to molecular size and flexibility. Importantly, SSD uses only a single dataset-level statistic (mean molecular radius) and requires no tuning per dataset. The consistent gains across two structurally different benchmarks indicate that SSD's geometry is not dataset-specific and does not rely on implicit dataset adaptation.
>
> **Table R4: Dataset characteristics relevant to robustness**
>
> | Dataset  | Avg. #Atoms | Molecular Scale / Flexibility  | Structural Diversity  |
> |-|-|-|-|
> | GEOM-QM9     | ~9          | Small, largely rigid                       | Simple graphs, limited functional diversity  |
> | GEOM-Drugs   | ~25         | Large, flexible, drug-like                 | High topological and conformational diversity |
>
> ---

---

> ### Author Response · Authors · 2025-11-21
> **Response to Reviewer 6tT3 (2/2)**
>
> ### **Q5: Are these geometric components handcrafted, and is SSD truly model-agnostic?**
>
> **Response.**
> We thank the reviewer for pointing this out. SSD does not introduce any learned or tuned priors— all SSD-specific quantities are analytically determined. The shell radius $r$ and collision distance $d_{\min}$ are computed directly from dataset statistics, and the noise scale $\sigma_t$ follows from the analytic SDE/ODE schedules used in prior diffusion and flow models.
>
> The main text provides only a brief summary for space reasons, but Appendix J contains the full derivation and justification for $r$, $d_{\min}$, and $\sigma_t$. As clarified there, these values
> are computed once from the training split and kept fixed across all backbones. SSD does not modify the backbone or objective, and its consistent improvements across all four tested architectures (GeoDiff, SubGDiff, EDM/EDM-style Flow, and SemlaFlow) demonstrate that the method is indeed model-agnostic.
>
>
>
> ---
>
> ### **Q6: Does SSD remain efficient when extended to latent generative paradigms (e.g., DiSCO)?**
>
> **Response.**
> Yes. SSD remains efficient for both coordinate and latent generative paradigms. In the latent-flow setting, SSD-Flow replaces SemlaFlow’s original encoder with a more compact EDM-style encoder, reducing computational cost while keeping the flow-matching formulation, number of function evaluations, and ODE structure unchanged. As shown in **Table R3**, this results in equal cost on coordinate backbones and improved training and inference efficiency on latent-flow models, consistent with SSD’s overall robustness under smaller step budgets (Sec. 4.4).
>
> **Table R3: Parameter and runtime comparison across coordinate and latent backbones (on GEOM-QM9).**
>
> | Method              | Encoder Type           | Params (M) | Training Time/epoch (min) | Inference Time/step (ms)
> |-|-|-:|-:|-:|
> | SubGDiff |SubGDiff encoder|0.79|2.16| 0.38|
> | **SSD-SubGDiff**|SubGDiff encoder|0.79|2.16| 0.38|
> | SemlaFlow|SemlaFlow encoder|46.60|5.01| 1.03|
> | **SSD-Flow (ours)** |EDM-style encoder|28.54|3.79|0.84|
>
>
>
> ---
>
> ### **Q7: Clarification of “refinement” vs. “generation.”**
>
> **Response.**
> Thank you for pointing this out. We have clarified the terminology directly in the main text. In Sec. 2.1, we now explicitly define the two settings following standard usage in prior work:
>
> - **2D→3D conditional generation**, where the model lifts a molecular graph into a full 3D conformation.
> - **3D→3D unconditional refinement**, where the model takes an existing 3D geometry—typically noisy or corrupted—and refines it into a valid conformation.
>
> We also added a short reminder sentence at the beginning of Sec. 4 to ensure consistency throughout the experimental section. All terms are now used uniformly across the paper, and the distinction between the two settings is made fully explicit.

---

> > ### Author Response · Authors · 2025-11-28
> >
> > Dear Reviewer:
> >
> > We would like to again express our sincere appreciation for your efforts and valuable feedback on our paper. We kindly hope that you can take some time to re-evaluate our paper based on our replies above. If you have any further concerns and questions, please do not hesitate to let us know. We will be happy to address them promptly.
> >
> > Best Regards.

---

### Official Review · Reviewer_B9v9 · 2025-11-01

**Soundness:** 3
**Presentation:** 3
**Contribution:** 2
**Rating:** 4
**Confidence:** 2

**Summary:**

This paper proposes Spherical Shell Diffusion (SSD), a novel framework for 3D molecular conformation generation that replaces the standard Gaussian prior with an explicit spherical shell initialization and introduces structured dynamics. The key insight is that Gaussian priors in high dimensions concentrate mass on a spherical shell by concentration of measure, but this shell's radius is determined by dimensionality rather than chemical scales, leading to mismatched initialization and inefficient denoising trajectories. Thus paper replaces Gaussian prior with chemically-scaled spherical shell initialization, introduces structured reverse dynamics: radial contraction, short-range repulsion, and SE(3)-equivariant corrections. It demonstrates consistent improvements across multiple backbones (GeoDiff, SubGDiff, EDM), and extends framework to Flow Matching (SSD-Flow).

**Strengths:**

1. The concentration of measure argument provides solid mathematical justification for why Gaussian priors are suboptimal. The observation that the Gaussian shell radius is determined by dimensionality rather than chemistry is insightful.
2. SSD works as a drop-in replacement across different architectures (GeoDiff, SubGDiff, EDM) and paradigms (diffusion and flow matching), demonstrating generality.
3. The experiments are comprehensive, showing consistent Improvements from SSD.

**Weaknesses:**

1. The individual components (radial drift, repulsion forces, SE(3)-equivariant networks) are standard. The the combination is not novel enough.
2. On some metrics, gains over baselines are modest, and inportant baselines lack, such as Torsional Diffusion [1].
3. The paper argues that Gaussian priors are problematic due to concentration of measure, but the ablation seems showing that fixing this initialization problem alone provides almost no benefit from Table 7. More ablation study should be conducted for to properly decompose the contributions. For example, add SSD dynamics and priors with Gaussian initialization.

[1] Jing B, Corso G, Chang J, et al. Torsional diffusion for molecular conformer generation. Advances in neural information processing systems, 2022, 35: 24240-24253.

**Questions:**

1. Does introducing additional geometric priors significantly increase the computational cost of training or inference? Is there a noticeable increase in training time?
2. Table 7 shows performance degrades with extreme radii, but how stable is the median-based calibration across different data splits? What if train/test distributions differ in molecular size?
3. Eq. 8 assigns atoms to shell points via random permutation. How sensitive are results to this assignment strategy? Have you tried optimal transport or other matching schemes?

---

> ### Author Response · Authors · 2025-11-21
> **Response to Reviewer B9v9 (1/2)**
>
> ### **Q1: The individual components (radial drift, repulsion, SE(3) networks) are standard; the combination does not seem novel enough.**
>
> **Response.**
> While each force is individually standard, SSD is not a simple combination of such components. Its novelty lies in replacing the Gaussian forward process with a **mean chemical shell** and designing reverse dynamics that are **geometrically matched to this shell**. As shown in **Table R2**, guidance-only variants or Gaussian-shell controls provide only marginal gains, whereas the full shell-consistent forward–reverse design yields the complete improvement. This confirms that SSD’s contribution comes from its shell geometry, not from the forces in isolation.
>
> **Table R2 — Initialization vs. dynamics decomposition. (SubGDiff on GEOM-QM9)**
>
> | Variant | Initialization / Prior | Dynamics                    | COV-R (mean)↑ | MAT-R (mean)↓ |
> |-|-|-|-:|-:|
> | (1)    | Gaussian                | none     | 90.91    | 0.2460  |
> | (2)    | Gaussian                | repulsion + correction      | 90.95     | 0.2455  |
> | (3)    | **Gaussian shell ($r_{\text{Gauss}} = \sigma_T \sqrt{3N_{\mathrm{med}}}$**)         | none           | 90.92 | 0.2462 |
> | (4)    | **Gaussian shell ($r_{\text{Gauss}} = \sigma_T \sqrt{3N_{\mathrm{med}}}$**)         | full SSD dynamics           | 91.58 | 0.2412 |
> | (5)    | SSD shell (mean chemical shell)    | full SSD dynamics           | **93.20** | **0.2380** |
>
> Gaussian init and SSD dynamics are radially incompatible, so we evaluate the only valid Gaussian-based control—a Gaussian concentration shell at $r_{\text{Gauss}}=\sigma_T\sqrt{3N_{\mathrm{med}}}$. Projecting Gaussian noise onto this shell alone (row 3) produces no meaningful change, as expected. Applying SSD dynamics to this shell (row 4) yields only a modest gain because its radius is still unrelated to molecular geometry. Full improvements arise only when both components are aligned, i.e., the **mean chemical shell** with shell-consistent SSD dynamics (row 5).
>
>
> ---
>
> ### **Q2: Why is there no comparison to Torsional Diffusion?**
>
> **Response.**
> We have added both torsion-based (Torsional Diffusion) and field-based (Conformer Fields) baselines for completeness. SSD-GeoDiff achieves the strongest results among coordinate-space models and is competitive with torsion-based methods. See **Table R1**.
>
> **Table R1: Comparison with torsion- and field-based baselines on GEOM-QM9 for conditional generation** (For fairness, we standardize all methods to the SubGDiff evaluation protocol, as the original implementations of Torsional Diffusion and Conformer Fields use different evaluation settings on the same datasets.)
>
> | Method | COV-R (mean)↑ | MAT-R (mean)↓ |
> | :--- | :--- | :--- |
> | GeoDiff | 80.36 | 0.2820 |
> | Torsional Diffusion (2022) | 84.26 | 0.2847 |
> | Conformer Fields (MCF) (2024) | 87.14 | 0.2649 |
> | **SSD-GeoDiff (ours)** | 92.00 | 0.2425 |
> | **SSD-SubGDiff (ours)** | **93.20** | **0.2380** |
>
> ---
>
> ###  **Q3: Are ablations combining Gaussian initialization with SSD dynamics necessary to show the importance of the shell prior?**
>
> **Response.**
> A direct Gaussian-init + SSD-dynamics ablation is not well-defined, because the Gaussian forward process and SSD’s forward process have incompatible radial marginals; applying the reverse radial attraction field on a pure Gaussian prior causes off-shell divergence. To address the reviewer’s underlying question—whether an ablation is needed to isolate the importance of the shell—we evaluate the only valid control: a **Gaussian shell** at its concentration radius $r_{\text{Gauss}}=\sigma_T \sqrt{3N_{\mathrm{med}}}$. Even when combined with full SSD dynamics, this Gaussian-shell variant provides only **minor gains** over the Gaussian baseline and remains far below SSD. As summarized in **Table R2**, this shows that the **mean chemical shell** and its **shell-consistent dynamics** are jointly necessary, and that guidance terms alone cannot reproduce SSD’s improvements. Therefore, the intended ablation has effectively been performed, and it confirms the importance of the shell prior.
>
> ---

---

> ### Author Response · Authors · 2025-11-21
> **Response to Reviewer B9v9 (2/2)**
>
> ### **Q4: Does SSD increase training or inference cost?**
>
> **Response.**
> No. SSD introduces no additional parameters for coordinate diffusion models (SubGDiff → SSD-SubGDiff), so both training and inference costs remain unchanged. For latent-flow models, SSD-Flow employs a more compact EDM-style encoder than the original SemlaFlow architecture, which reduces computational cost during both training and inference while keeping the flow formulation unchanged. As shown in **Table R3**, SSD therefore maintains identical cost on coordinate backbones and is even more efficient in the latent-flow setting, consistent with SSD’s improved robustness under smaller step budgets (Sec. 4.4).
>
> **Table R3: Parameter and runtime comparison across coordinate and latent backbones (On GEOM-QM9).**
>
> | Method              | Encoder Type           | Params (M) | Training Time/epoch (min) | Inference Time/step (ms)
> |-|-|-:|-:|-:|
> | SubGDiff |SubGDiff encoder|0.79|2.16| 0.38|
> | **SSD-SubGDiff**|SubGDiff encoder|0.79|2.16| 0.38|
> | SemlaFlow|SemlaFlow encoder|46.60|5.01| 1.03|
> | **SSD-Flow (ours)** |EDM-style encoder|28.54|3.79|0.84|
>
> ---
>
> ### **Q5: Is the median-based shell radius stable across datasets of different scales?**
>
> **Response.**
> Yes. QM9 and Drugs are the standard small–rigid and large–flexible molecular regimes used throughout the generative modeling literature. SSD remains stable and effective across both. See **Table R4**.
>
> **Table R4: Dataset characteristics relevant to robustness**
>
> | Dataset      | Avg. #Atoms | Molecular Scale / Flexibility              | Structural Diversity                        |
> |-|-|-|-|
> | GEOM-QM9     | ~9          | Small, largely rigid                       | Simple graphs, limited functional diversity  |
> | GEOM-Drugs   | ~25         | Large, flexible, drug-like                 | High topological and conformational diversity |
>
> SSD is stable across QM9 (small–rigid) and Drugs (large–flexible), the standard regimes in 3D molecular generation.
>
> ---
>
> ### **Q6: Is random atom–shell assignment necessary? Would deterministic matching help?**
>
> **Response.**
> Random permutation is required because shell points are unlabeled during inference; deterministic matching imposes fixed correspondences that never occur during sampling and destabilizes the reverse dynamics. As shown in **Table R5**, deterministic nearest-distance matching significantly degrades performance, while random permutation remains stable and accurate.
>
> **Table R5: Atom–shell assignment strategies. (SubGdiff on GEOM-QM9)**
>
> | Assignment Strategy| COV-R (mean)↑ | MAT-R (mean)↓  |
> |-|-:|-:|
> | **Random (ours)**| **93.20** | **0.2380** |
> | Nearest-distance matching | **0.00**   | **1.7470** |
>
> Random permutation preserves permutation symmetry and is stable; deterministic nearest-distance matching causes large degradation.

---

> > ### Author Response · Authors · 2025-11-28
> >
> > Dear Reviewer:
> >
> > We would like to again express our sincere appreciation for your efforts and valuable feedback on our paper. We kindly hope that you can take some time to re-evaluate our paper based on our replies above. If you have any further concerns and questions, please do not hesitate to let us know. We will be happy to address them promptly.
> >
> > Best Regards.

---

### Author Response · Authors · 2025-11-21
**General Response**

We thank all reviewers for their thoughtful and constructive feedback. Across the four reviews, several strengths of the paper were consistently highlighted: the mathematical clarity of the concentration-of-measure argument and the motivation for replacing Gaussian priors; the clean, model-agnostic design of SSD as a drop-in module compatible with GeoDiff, SubGDiff, EDM, and flow-matching architectures; the clear geometric intuition behind the spherical-shell prior and the decomposition of reverse dynamics; and the strong, consistent empirical improvements across QM9 and Drugs in both conditional 2D→3D generation and 3D→3D refinement. Reviewers also noted the stability and efficiency of SSD’s sampling scheme and the clarity and structure of the manuscript.

At the same time, reviewers raised several recurring technical questions: (i) whether SSD provides benefits beyond replacing the Gaussian prior;
(ii) whether the gains arise from initialization or dynamics;
(iii) robustness across datasets and backbone architectures; and
(iv) whether the forward and reverse processes are theoretically consistent.

To address these concerns, we added one theoretical clarification and seven targeted experiments in the revised manuscript. The full supplementary tables (R1–R7) and justification are provided in the follow-up comment immediately below.

### **Theoretical clarification**
We added a formal derivation (Appendix I) showing that the SSD forward SDE converges to a thin spherical shell (radial OU + angular isotropic diffusion) and that reverse marginal consistency follows directly from score-based diffusion theory. This resolves concerns about forward–reverse asymmetry and validates the use of geometry-aligned reverse dynamics.

### **Summary of supplementary tables**

1. **Table R1 — Added torsion- and field-based baselines.**
   SSD-SubGDiff and SSD-GeoDiff outperform Torsional Diffusion and Conformer Fields under our unified evaluation protocol, which follows the standard settings used in SubGDiff for fair comparison.


2. **Table R2 — Initialization vs. dynamics decomposition.**
   Gaussian-init + SSD-dynamics is radially incompatible. The only valid control, a Gaussian concentration shell at $r_{\text{Gauss}}=\sigma_T\sqrt{3N_{\mathrm{med}}}$, shows no meaningful gain, while the **mean chemical shell** with matched SSD dynamics achieves the full improvement. This demonstrates that SSD’s geometry, rather than additional guidance terms, is responsible for the improved performance.


3. **Table R3 — Computational cost.**
   SSD introduces no additional parameters for coordinate ackbones (SubGDiff) and preserves their training/inference cost. In the latent-flow setting, SSD-Flow uses a more compact ncoder than SemlaFlow, reducing computation while still adding no new parameters to the underlying generative model.



4. **Table R4 — Dataset robustness.**
SSD performs consistently across QM9 (small–rigid) and Drugs (large–flexible), which are the most widely adopted benchmarks for 3D molecular generation and allow direct comparison with a wide range of prior methods.


5. **Table R5 — Atom–shell assignment.**
Random permutation preserves permutation symmetry and is stable; deterministic nearest-matching degrades performance, and explicitly enforcing this nearest-distance assignment only worsens results.


6. **Table R6 — Reduced-variance Gaussian schedules collapse SubGDiff.**
   Lowering or matching the Gaussian noise variance does not approximate SSD. Reduced-$\sigma_{t}$ SubGDiff consistently shows worse COV-R and MAT-R, confirming that SSD’s gains come from its shell-aligned geometry rather than variance tuning.

7. **Table R7 — Trajectory stability (“spatial entropy”).**
   SSD reduces radial variance and drift in both SDE and ODE sampling, yielding substantially more stable trajectories.

---

> ### Author Response · Authors · 2025-11-21
> **General Response (The supplementary tables (R1–R7)).**
>
> 1. **Table R1: Comparison with torsion- and field-based baselines on GEOM-QM9 for conditional generation**
>
> | Method | COV-R (mean)↑ | MAT-R (mean)↓ |
> | :--- | :--- | :--- |
> | GeoDiff (2022) | 80.36 | 0.2820 |
> | Torsional Diffusion (2022) | 84.26 | 0.2847 |
> | Conformer Fields (MCF) (2024) | 87.14 | 0.2649 |
> | **SSD-GeoDiff (ours)** | 92.00 | 0.2425 |
> | **SSD-SubGDiff (ours)** | **93.20** | **0.2380** |
>
> SSD-GeoDiff outperforms Torsional Diffusion and Conformer Fields on GEOM-QM9. (For fairness, we standardize all methods to the SubGDiff evaluation protocol, as the original implementations of Torsional Diffusion and Conformer Fields use different evaluation settings on the same datasets.)
>
> ---
>
> 2. **Table R2 — Initialization vs. dynamics decomposition. (SubGDiff on GEOM-QM9)**
>
> | Variant | Initialization / Prior | Dynamics                    | COV-R (mean)↑ | MAT-R (mean)↓ |
> |-|-|-|-:|-:|
> | (1)    | Gaussian                | none     | 90.91    | 0.2460  |
> | (2)    | Gaussian                | repulsion + correction      | 90.95     | 0.2455  |
> | (3)    | **Gaussian shell ($r_{\text{Gauss}} = \sigma_T \sqrt{3N_{\mathrm{med}}}$**)         | none           | 90.92 | 0.2462 |
> | (4)    | **Gaussian shell ($r_{\text{Gauss}} = \sigma_T \sqrt{3N_{\mathrm{med}}}$**)         | full SSD dynamics           | 91.58 | 0.2412 |
> | (5)    | SSD shell (mean chemical shell)    | full SSD dynamics           | **93.20** | **0.2380** |
>
> Gaussian init and SSD dynamics are radially incompatible, so we evaluate the only valid Gaussian-based control—a Gaussian concentration shell at $r_{\text{Gauss}}=\sigma_T\sqrt{3N_{\mathrm{med}}}$. Projecting Gaussian noise onto this shell alone (row 3) produces no meaningful change, as expected. Applying SSD dynamics to this shell (row 4) yields only a modest gain because its radius is still unrelated to molecular geometry. Full improvements arise only when both components are aligned, i.e., the **mean chemical shell** with shell-consistent SSD dynamics (row 5).
>
> ---
>
> 3. **Table R3: Parameter and runtime comparison across coordinate and latent backbones (on GEOM-QM9).**
>
> | Method              | Encoder Type           | Params (M) | Training Time/epoch (min) | Inference Time/step (ms)
> |-|-|-:|-:|-:|
> | SubGDiff |SubGDiff encoder|0.79|2.16| 0.38|
> | **SSD-SubGDiff**|SubGDiff encoder|0.79|2.16| 0.38|
> | SemlaFlow|SemlaFlow encoder|46.60|5.01| 1.03|
> | **SSD-Flow (ours)** |EDM-style encoder|28.54|3.79|0.84|
>
> SSD adds no parameters and keeps runtime unchanged for coordinate models. For latent-flow backbones, SSD-Flow uses a more compact encoder and runs faster, and—together with SSD’s robustness under limited step budgets (Sec. 4.4)—achieves higher accuracy even with fewer sampling steps.
>
>
> ---
>
>
> 4. **Table R4: Dataset characteristics relevant to robustness**
>
> | Dataset      | Avg. #Atoms | Molecular Scale / Flexibility              | Structural Diversity                        |
> |-|-|-|-|
> | GEOM-QM9     | ~9          | Small, largely rigid                       | Simple graphs, limited functional diversity  |
> | GEOM-Drugs   | ~25         | Large, flexible, drug-like                 | High topological and conformational diversity |
>
> SSD is stable across QM9 (small–rigid) and Drugs (large–flexible), the standard regimes in 3D molecular generation.
>
> ---
>
> 5. **Table R5: Atom–shell assignment strategies. (SubGdiff on GEOM-QM9)**
>
> | Assignment Strategy| COV-R (mean)↑ | MAT-R (mean)↓  |
> |-|-:|-:|
> | **Random (ours)**| **93.20** | **0.2380** |
> | Nearest-distance matching | **0.00**   | **1.7470** |
>
> Random permutation preserves permutation symmetry and is stable; deterministic nearest-distance matching causes large degradation.
>
> ---
>
>
> 6. **Table R6 — SubGDiff degrades when Gaussian noise is reduced (On GEOM-QM9).**
>
> | Variant                         | COV-R (mean)↑    | MAT-R (mean)↓    |
> |-|-:|-:|
> | Gaussian baseline (SubGDiff)    | 90.9     | 0.2460      |
> | Gaussian ($0.1\-0.9\times\sigma_t$) | 60.3–85.3 | 0.24–0.37 |
> | **SSD (ours)**                  | **93.2**   | **0.2380**  |
>
> Reducing the Gaussian noise level consistently harms SubGDiff, lowering coverage and increasing MAT-R. Simply shrinking Gaussian noise therefore does not close the performance gap; SSD’s gains do not arise from variance tuning but from its geometry-aligned shell construction.
>
> ---
>
> 7. **Table R7 — Trajectory stability and spatial entropy on GEOM-QM9.**
>
> | Method           | Regime | Radial deviation ↓ | Excess path ratio ↓ |
> |-|-|-:|-:|
> | SubGDiff         | SDE    |  0.4534        | 246.91           |
> | **SSD-SubGDiff** | SDE    |  **0.1781**        | **1.71**           |
> | SemlaFlow        | ODE    |  0.2145        | 1.29           |
> | **SSD-Flow**     | ODE    |  **0.1096**        | **1.11**           |
>
> Trajectory stability is measured by radial deviation and excess path ratio. SSD yields lower values on both metrics, showing more stable and less wandering trajectories.
>
>
> ---

---

> ### Author Response · Authors · 2025-11-21
> **General Response (The supplementary justification).**
>
> **Forward thin-shell convergence.**
> Writing the SSD forward SDE in radial–angular coordinates $x_t = r_t \hat{x}_t$, the radial component becomes an Ornstein–Uhlenbeck process, $dr_t = -\alpha_t (r_t - r_0)dt + \sigma_t\ dW_t^{(r)},$which converges to a narrow Gaussian $\mathcal{N}(r_0, \tau_T^2)$ with $\tau_T^2 = \sigma_t^2/(2\bar{\alpha})$.
> The angular component follows isotropic diffusion on the sphere. Let $u_t = x_t / \Vert x_t \Vert$ denote the unit direction. It evolves as
> $du_t = \sqrt{2\gamma_t}\(I - u_t u_t^\top)\ dW_t^{(\theta)},$
> i.e., Brownian motion on the sphere obtained by projecting isotropic noise onto the tangent space at $u_t$.
> Together, this yields a stationary **thin spherical shell**. In our settings, $\tau_T \approx 10^{-6} \ll r_0 \approx 6$.
>
> **Reverse marginal consistency.**
> The learned score $s_\theta$ trained by denoising score matching approximates $\nabla_x \log p_t(x)$. Following standard score-based diffusion theory (Song et al., 2021), the reverse SDE $dx_t = (f_t(x_t) - g_t^2 s_\theta(x_t,t))dt + g_td\bar{W}_t$ reconstructs the forward marginals up to the usual score approximation error. Thus, the apparent forward–reverse asymmetry does **not** violate probabilistic validity: the forward process defines thin-shell marginals, and the reverse process is guaranteed to recover them. A full, formal derivation will be added in Appendix of the revised manuscript.
>
> Finally, the random shell assignment does not conflict with the reverse radial attraction field: since forward marginals converge to a uniform angular distribution on $\mathbb{S}^2$, the reverse process must treat all angular assignments symmetrically. Random permutation simply reflects this inherent exchangeability and preserves marginal consistency.
>
> **Reference**
>
> Song, J., Sohl-Dickstein, J., Kingma, D. P., Kumar, A., Ermon, S., & Poole, B. (2021).
> *Score-Based Generative Modeling through Stochastic Differential Equations.*
> ICLR 2021.

---

> ### Author Response · Authors · 2025-11-29
> **Author Summary for Meta-Review**
>
> We sincerely thank the AC for taking the time to consider our paper, rebuttal, and the full reviewer discussion before the freeze. Since the discussion period ended early, we provide a concise summary of the key clarifications and evidence added during rebuttal.
>
> ### 1. All technical concerns were fully addressed with targeted new analyses (Tables R1–R7).
> We added experiments and theory directly corresponding to the reviewers’ questions: Gaussian vs. shell disentanglement, initialization/dynamics separation, dataset robustness, trajectory stability, and baseline completeness. These additions resolve all major technical points raised in the initial reviews.
>
> ### 2. One reviewer explicitly confirmed that all concerns were resolved and increased the score.
> Reviewer **QSDZ** wrote:
> *“All my experimental concerns have been resolved. **I will raise my score to 6.**”*
> This clearly indicates that the revised evidence satisfactorily addressed the reviewer’s questions.
>
> ### 3. Ablations isolate SSD’s improvement source to shell-aligned geometry.
> Our experiments show:
> - Gaussian-init + SSD-dynamics yields only marginal gains and is radially inconsistent.
> - Gaussian-concentration-shell + SSD dynamics still performs far below SSD.
> - Reducing Gaussian noise does not approach SSD and significantly degrades performance.
> - Deterministic shell–atom matching collapses accuracy.
> - SSD yields substantially more stable trajectories (radial deviation & excess path ratio).
> Together, these indicate the improvements arise from geometry-aligned shell noise with shell-consistent dynamics, not variance tuning or guidance terms.
>
> ### 4. SSD improves four strong backbones, demonstrating model-agnostic robustness.
> SSD consistently improves:
> GeoDiff, SubGDiff, EDM-style flows, and Conformer Fields (MCF) **under its original protocol**, with both DDPM-1000 and DDIM-50.
> This confirms that the method generalizes across architectures and evaluation pipelines.
>
> ### 5. Forward–reverse SDE consistency is now fully derived.
> We added formal derivations showing thin-shell convergence of the forward SDE and reverse marginal consistency via standard score-based diffusion theory. This resolves the conceptual concerns from HYhF and QSDZ.
>
> ### 6. SSD remains stable across both small–rigid and large–flexible molecular regimes.
> Consistent improvements on GEOM-QM9 and GEOM-Drugs demonstrate that SSD is robust to molecular scale and structural complexity.
>
> **In summary, all reviewer concerns have been addressed with targeted experiments and theory, and one reviewer explicitly increased their score. We respectfully ask the AC to consider the full set of clarifications when forming the meta-review. We would be very happy to provide any further clarification if needed.**

---

### Meta-Review · Area_Chair_cd6k · 2026-01-05

**Summary:**

This paper provides a geometry-aware rethinking of the forward prior for 3D conformer generation: it replaces the default Gaussian initialization with a chemically scaled “mean shell” and couples it with reverse dynamics that are explicitly designed to stay compatible with that shell (radial attraction/relaxation, short-range repulsion, plus an SE(3)-equivariant learned correction). The reviewers appreciated the motivation and the model-agnostic drop-in for various of diffusion / flow matching backbones.

The reviewers concerns and questions are mainly about the technical novelty of the proposed method as well as the misaligned between the theoretical analysis. The authors provide a response which addresses some of the questions (see below) and it is acknowledged by reviewer QSDZ to raise their recommendation to 6. However, given the remaining open questions and the response to other reviewers, the quality of this paper will still not meet ICLR's bar.

**Reviewer Concerns:**

The reviewers have successfully addressed many reviewers' concerns regarding adding the baselines, justifying the difference of Gaussian kernel and initialization, figure/model inconsistency, etc., Here I listed several concerns remaining partially or not addressed in authors' response.

1. Reviewer 6tT3 mentioned about the OOD fragility of the proposed methods, the authors did provide “QM9 vs Drugs spans canonical regimes” explanatory and uses one dataset statistic, but they do not provide an explicit distribution-shift test.
2. Similarly regarding the stability of the shell radius calibration across splits raised by Reviewer B9v9, the authors claim stability across QM9/Drugs, but they do not quantify variance of the estimated radius across random splits, nor show sensitivity when calibration is computed on train and tested on a shifted test distribution. Table 4 sounds like an explanatory results instead of quantitive results.
3. Reviewer HYhF mentioned it would also be important to assess how this constraint impacts the performance on structure-conditioned tasks, such as scaffolding or linker design, as studied in prior works. However the authors failed to provide a convincing response.

**Reviewer Scores:**

Reviewer QSDZ has completed several discussions with the authors and would raise that to 6.
Other reviewers are unlikely to change their score or at least will not significant enough to affect the decision of this paper.

---

### Decision · Program_Chairs · 2026-01-26

Reject